# Signatures of quantum spin liquid state and unconventional transport in thin film TbInO₃

Johanna Nordlander [1,2,3,14] ✉, Margaret A. Anderson [1,14], Tony Chiang [4], Austin Kaczmarek [5], Nabaraj Pokhrel[6], Kuntal Talit [7], Spencer Doyle[1], Edward Mercer [8,9], Christian Tzschaschel [10], Jun-Ho Son[11], Hesham El-Sherif[12], Charles M. Brooks[1], Eun-Ah Kim[11], Alberto de la Torre [8,9], Ismail El Baggari[12], Elizabeth A. Nowadnick [7], Katja C. Nowack [5], John T. Heron [4] & Julia A. Mundy [1,13] ✉

Quantum spin liquids, where the frustrated magnetic ground state hosts highly entangled spins resisting long-range order to 0 K, are exotic quantum magnets proximate to unconventional superconductivity and candidate platforms for topological quantum computing. Although several quantum spin liquid material candidates have been identified, thin films crucial for device fabrication and further tuning of properties remain elusive. Recently, hexagonal TbInO₃ has emerged as a quantum spin liquid candidate which also hosts improper ferroelectricity and exotic high-temperature carrier transport. Here, we synthesize thin films of TbInO₃ and characterize their magnetic and electronic properties. Our films present a highly frustrated magnetic ground state without long-range order to 0.4 K, consistent with bulk crystals. We further reveal a rich ferroelectric domain structure and unconventional non-local transport near room temperature, suggesting hexagonal TbInO₃ as a promising candidate for realizing exotic magnetic and transport phenomena in epitaxial heterostructures.

One of the most exotic ground states in condensed matter physics, a spin liquid is a state of matter that lacks a unique lowest energy spin configuration and instead hosts a macroscopically degenerate, highly correlated ground state. Such a state is characterized by the fluctuation of spins between the degenerate spin configurations and is the consequence of magnetic frustration, which develops when competing magnetic interactions of neighboring spins are prevented from being simultaneously satisfied[1]. In the classical regime, the spins in a spin liquid thermally fluctuate as long as the thermal energy exceeds the barrier between states, and, upon cooling below a certain temperature, the spins tend to either develop long-range order or freeze into a spin glass state[2]. In a quantum spin liquid, on the other hand, the magnetic ground state is a superposition of all the lowest energy spin configurations and the entangled spins continue to fluctuate down to a temperature of absolute zero due to the existence of quantum fluctuations even in the absence of thermal fluctuations[2,3]. Such states are not only a physical curiosity but also may hold insight into other condensed matter phenomena. It has been suggested that the

[1]Department of Physics, Harvard University, Cambridge, MA, USA. [2]Paul Drude Institute for Solid State Electronics, Berlin, Germany. [3]Department of Physics, University of Zurich, Zurich, Switzerland. [4]Department of Materials Science and Engineering, University of Michigan, Ann Arbor, MI, USA. [5]Department of Physics, Laboratory of Atomic and Solid State Physics, Cornell University, Ithaca, NY, USA. [6]Department of Physics, University of California, Merced, CA, USA. [7]Department of Chemical and Materials Engineering, University of California, Merced, CA, USA. [8]Department of Physics, Northeastern University, Boston, MA, USA. [9]Quantum Materials and Sensing Institute, Northeastern University, Burlington, MA, USA. [10]Department of Chemistry and Chemical Biology, Harvard University, Cambridge, MA, USA. [11]Department of Physics, Cornell University, Ithaca, NY, USA. [12]Rowland Institute, Harvard University, Cambridge, MA, USA. [13]School of Engineering and Applied Sciences, Harvard University, Cambridge, MA, USA. [14]These authors contributed equally: Johanna Nordlander, Margaret A. Anderson. ✉e-mail: johanna.nordlander@physik.uzh.ch; mundy@fas.harvard.edu

resonating valence bond state of such a phase may be proximate to high-temperature superconductivity, as realized in the cuprates[4]. In addition, the spinon quasiparticle excitations of a quantum spin liquid can be itinerant Majorana fermions with a gapless dispersion, suggesting that these materials could be a possible platform for topological quantum computing[3,5].

Despite longstanding interest in quantum spin liquids, there have been limited materials systems identified that host an intrinsic quantum spin liquid ground state. Even one of the most studied candidates, $RuCl_3$, orders at low-temperature and is only proposed to reach a spin liquid state at high magnetic fields[6–8]. Recently, hexagonal $TbInO_3$ has emerged as a promising new candidate to realize the putative Kitaev spin liquid[9,10]. In $TbInO_3$, quasi two-dimensional layers of $Tb^{3+}$ ions in a triangular configuration are separated by layers of non-magnetic $InO_5$ polyhedra (Fig. 1a–c), leading to strongly frustrated magnetic interactions on the isolated terbium sublattice[11–13] that lacks magnetic ordering to 0.1 K[9]. The material undergoes a structural transition at high-temperature: a coordinated tilting of the $InO_5$ trigonal bipyramids and an accompanying distortion of the terbium layers breaks inversion symmetry to generate an improper ferroelectric[10,14–17]. This trimerization of the lattice leads to a unit-cell tripling with two inequivalent terbium sites, Tb1 and Tb2, and it has been proposed that the resulting honeycomb sublattice (Fig. 1c) is magnetically isolated from the remainder of the terbium atoms[9,10]. Further evidence from μSR measurements suggest dynamic fluctuations of the terbium moments on the honeycomb sites to the lowest temperatures and inelastic neutron diffraction proposes a gapless spin structure[9]. Most intriguingly, recent optical measurements have suggested that the exotic excitations characteristic of this highly entangled quantum ground state may persist to room-temperature[18], positing $TbInO_3$ as a leading candidate to probe not only the low-temperature quantum spin liquid state but also new emergent physics at higher temperatures.

Despite this considerable interest in both the low and high-temperature properties of $TbInO_3$, to date, this material, like many quantum spin liquid candidates, has only been synthesized as a single crystal and in powder form. While such large samples are amenable to canonical characterization methods, such as inelastic neutron scattering and thermal transport[8], the realization of a spin liquid in the thin film form would enable novel methods to tune the materials ground state through epitaxial strain and heterostructuring[19] as well as a platform to explore non-local properties in a device-ready geometry.

Here, we report thin-film synthesis of the quantum spin liquid candidate hexagonal $TbInO_3$. We reveal an absence of long-range magnetic order down to at least 0.4 K and an absence of any type of spin freezing or disorder effects down to at least 1 K, indicative of strong magnetic frustration and demonstrating that the ground state of the bulk crystals is highly preserved in the thin film limit. Using non-local transport measurements based on the inverse spin Hall effect, we further reveal unconventional carrier transport in $TbInO_3$ films at temperatures well above the conventional quantum spin liquid regime. We find that this behavior is not consistent with either ferromagnetic or antiferromagnetic order and thus points to novel exotic physics potentially connected to the entangled quantum ground state observed by optical measurements of $TbInO_3$ crystals[18].

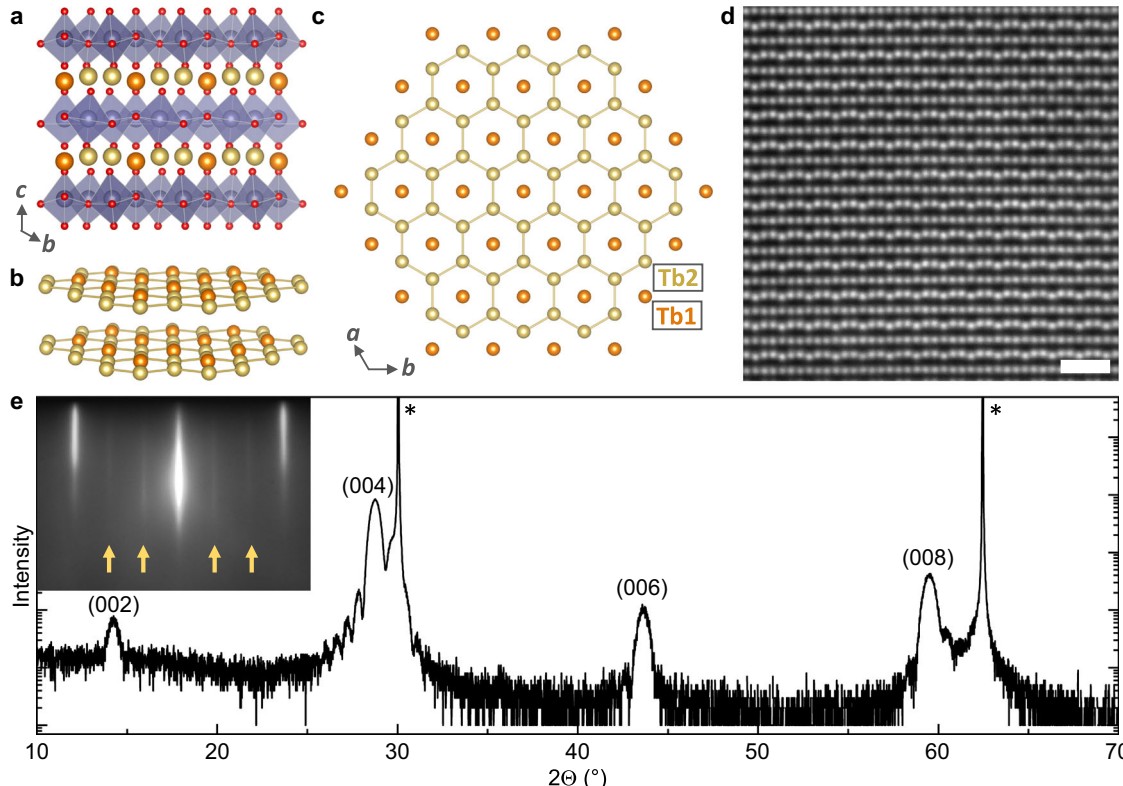

**Fig. 1 | $TbInO_3$ structure and thin-film growth. a** Crystal structure of $TbInO_3$ in the hexagonal $P6_3cm$ ferroelectric phase. The terbium ions (orange and gold for Tb1 and Tb2 sites, respectively) form a quasi-2-dimensional lattice, as visualized in (**b**), sandwiched between nonmagnetic $InO_5$ trigonal bipyramid layers (purple, with oxygen in red). **c** The ferroelectric distortion causes a stuffed honeycomb geometry in the Tb sublattice. **d** HAADF-STEM image acquired along $TbInO_3$ [100], showing the layered hexagonal phase of an epitaxial $TbInO_3$ thin film grown on YSZ(111). The corrugated Tb layers reveal the ferroelectric nature of the film. The scale bar is 1 nm. **e** X-ray diffraction $\theta−2\theta$ scan showing a stoichiometric hexagonal $TbInO_3$ film on YSZ without impurity phases. The asterisks indicate substrate diffraction peaks. Inset: RHEED image taken along $TbInO_3$ [100] after thin-film growth. The ferroelectric lattice trimerization is seen directly at the deposition temperature of 860 °C as a tripling of the unit cell leading to 1/3 order streaks as indicated by the arrows.

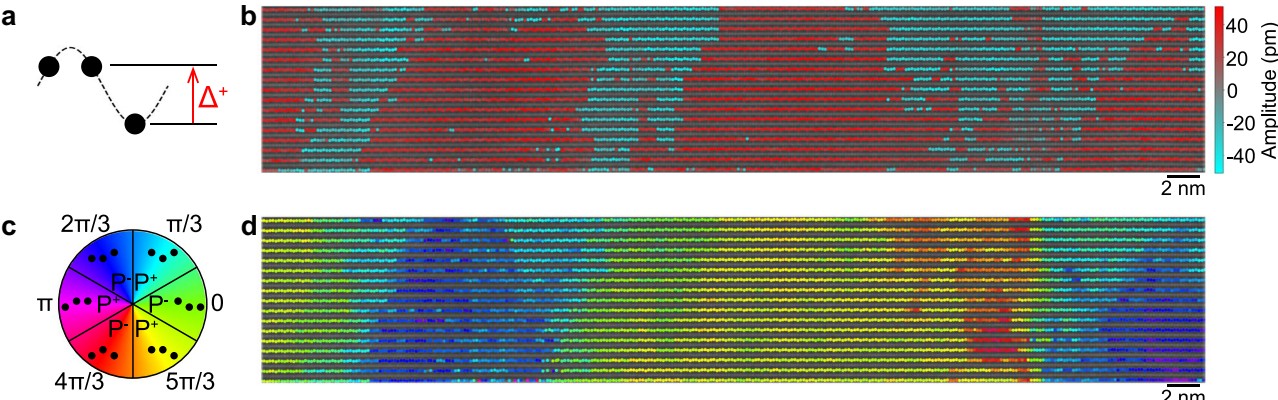

**Fig. 2 | Improper ferroelectric domain mapping in a 15 nm TbInO₃ epitaxial thin film. a** The up-up-down trimerization of the terbium sublattice fit to a sinusoid to extract the distortion amplitude (Δ) and phase. **b** A map of the local distortion amplitude in a 15 nm TbInO₃ thin film overlaid on a HAADF-STEM micrograph. **c** The trimerization phase corresponds to six possible ferroelectric domains. **d** A map of the improper ferroelectric domains in the same region as (**b**), based on the color scale given in (**c**).

## Results

### Thin-film growth

Among the group of quantum spin liquid candidate materials, TbInO₃ lends itself particularly well to epitaxial thin-film realization: it is well lattice matched to commercially available substrates and belongs to a larger family of isostructural functional hexagonal oxides enabling further epitaxial engineering and integration into oxide electronic heterostructures[20]. Here, we use reactive oxide molecular-beam epitaxy (MBE) to synthesize ultrathin epitaxial films of TbInO₃ on (111)-oriented yttria-stabilized zirconia (YSZ) substrates. The growth was optimized with respect to the Tb/In stoichiometry using in situ reflection high-energy electron diffraction (RHEED), see Supplementary Fig. S1, and confirmed with ex situ X-ray diffraction (XRD) and scanning transmission electron microscopy (STEM). A narrow growth window is identified for phase-pure TbInO₃ films (Supplementary Figs. S1 and S2)[21]. The $(\sqrt{3} \times \sqrt{3})$ R30° reconstruction seen in the RHEED pattern of stoichiometric TbInO₃ is indicative of the lattice trimerization, which serves as the primary order parameter of the improper ferroelectricity. This lattice trimerization is observed directly at the growth temperature and confirms a high ferroelectric $T_C$ exceeding our growth temperature of 860 °C. Figure 1e shows a $\theta$–$2\theta$ XRD scan, which confirms the purely (001)-oriented layered hexagonal structure. Laue oscillations around the (004) reflection of TbInO₃ are testament to sharp film interfaces, as further confirmed by a low surface roughness determined by atomic-force microscopy (Supplementary Fig. S1). The in-plane lattice mismatch between bulk TbInO₃(0001) and a 30° rotation of YSZ(111) is −0.8%. In-plane X-ray reciprocal space mapping (RSM) reveals that the films gradually relax with respect to the substrate lattice with increasing thickness (Supplementary Fig. S3). While at a thickness of 8 nm, the film lattice is fully aligned with the substrate, the 15 nm and 29 nm films show gradual relaxation. Likewise, local strain maps based on STEM images show that the 8 nm film has a uniform $\epsilon_{xx}$ across the interface (Supplementary Fig. S4). In the 15 nm film, except for a region of increased $\epsilon_{xx}$ near the film-substrate interface, there is a slight negative $\epsilon_{xx}$ within the film (Supplementary Fig. S5), which reveals partial relaxation of the in-plane strain, as expected from the X-ray RSM data.

We use STEM to visualize the atomic structure of the film (Fig. 1d and Supplementary Fig. S6). As shown in the high-angle annular dark field STEM (HAADF-STEM) image in Fig. 1d, the film is highly crystalline and ordered, composed of alternating layers of terbium and indium atoms. This layered structure confines the spins on the terbium sublattice to 2-D planes, providing a frustrated environment[9].

The terbium sublattice further displays the up-up-down displacement pattern, which is a consequence of the trimerization of the lattice structure (Fig. 2a). This lattice distortion drives the ferroelectric order in hexagonal indates, just as in hexagonal manganites[14,15] and ferrites[22]. By fitting the corrugation of the terbium lattice, both the amplitude and phase of the order parameter can be spatially resolved[23]. The trimerization mapping shown in Fig. 2b–d reveals a multi-domain state with a preference of up-polarized domains at the film-substrate interface, interspersed with down-polarized domains that tend to widen towards the top surface of the film. This is in contrast to the preferably single-domain state found in both hexagonal manganites and ferrites grown on the same type of YSZ substrate[24,25]. A similar nanometer-scale multi-domain state is realized in YMnO₃ thin films on platinum-coated Al₂O₃ substrates[26,27]. The TbInO₃ thin films display all six trimerization domains (Fig. 2b–d and Supplementary Figs. S7 and S8) with a smaller domain size and thus larger number of domain walls in comparison to the bulk crystals of TbInO₃[10]. A thinner 8 nm TbInO₃ also shows a similar domain pattern (Supplementary Fig. S7). It has been proposed that the domain walls could host magnetic edge states or novel spin excitations[10], making the TbInO₃ thin films with rich domain structure[23] an exciting platform to study these emergent properties.

We extract the average Tb1–Tb2 displacement amplitude (Fig. 2a) to be 34.8 pm, which is slightly less than the value of ~40 pm reported for bulk crystals[10]. Using density functional theory (DFT), we estimate the magnitude of the spontaneous polarization in our films, given a 35 pm Tb1–Tb2 displacement, to be 7.11 μC cm⁻² (See Supplementary Note I). The thinner sample shows a slightly smaller displacement (Supplementary Fig. S8), possibly due to strain or interface clamping imparted from the substrate. While the displacement in the thicker film corresponds to a polarization similar to the reported value of the ferroelectric polarization of the hexagonal manganites and ferrites, our estimated polarization is larger than the electrical polarization measured in bulk TbInO₃[10] at 77 K, despite our slightly smaller distortion.

### Frustrated rare-earth magnetism

Having established the growth of high-quality epitaxial thin films, we next investigate the low-temperature magnetic properties of our TbInO₃ films. Bulk crystal TbInO₃ has been suggested to host a quantum spin liquid ground state characterized by persistent spin fluctuations down to at least 0.1 K in the absence of long-range magnetic order[9,10]. It remains unclear, however, how such a state transfers to the

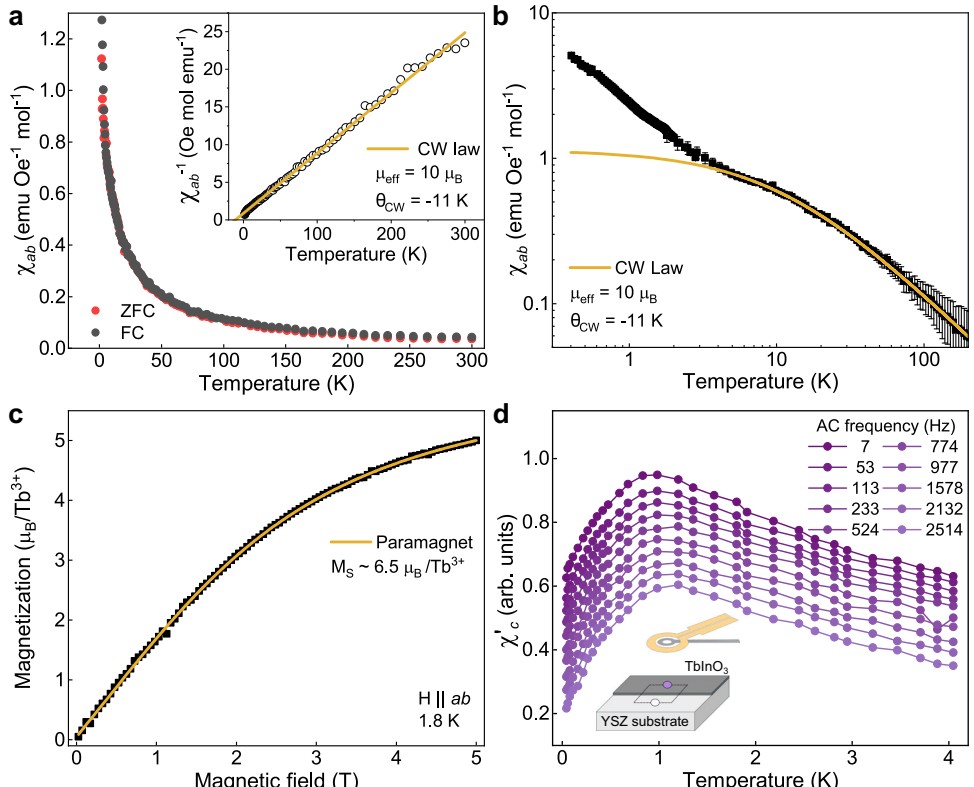

**Fig. 3 | SQUID magnetometry on a 38 nm film of TbInO₃ on YSZ(111). a** In-plane magnetic susceptibility measured using an applied field of 1000 Oe as a function of temperature upon warming after field-cooling (black) and zero-field cooling (red). No splitting of the two curves, as would be indicative of magnetic ordering, is observed down to 1.8 K. A linear Curie−Weiss fit to the inverse susceptibility yields a Curie−Weiss temperature of −11 K and an effective $Tb^{3+}$ moment $\mu_{eff}$ of approx. 10 $\mu_B$. **b** Magnetic susceptibility extended further down to 0.4 K, using a field of 2000 Oe, similarly reveals no long-range order. A deviation from the high-temperature Curie−Weiss behavior is evident below 5 K. The error bars represent the measurement uncertainty as provided by the instrument (see "Methods"). **c** In-plane magnetic field dependence of magnetization yields a saturation magnetization $M_S$ of ca. 6.5 $\mu_B$/$Tb^{3+}$ at 1.8 K. **d** AC magnetic susceptibility along the out-of-plane $c$-axis of the film, collected between 44 mK and 4 K by scanning SQUID magnetometry, displays a sharp downturn below 1 K. A background contribution from the substrate has been characterized on an area where the film had been removed (see schematic in inset) and subtracted from the data. The $y$-axis corresponds to the 7 Hz dataset (in emu mol⁻¹), all other curves have been offset for clarity.

thin-film limit. Here, we use a combination of conventional and scanning SQUID magnetometry to gain insight into the nature of the magnetic ground state of our epitaxial films.

Firstly, we note that the YSZ substrates used here, even though intrinsically diamagnetic, are found to exhibit a weak paramagnetic response, due to trace amounts of magnetic impurities. Hence, low-temperature measurements, where such paramagnetic substrate contributions become larger, are particularly challenging given the small magnetic volume of the thin film. We account for this parasitic magnetic signal by measuring both the magnetic signal of the total film + substrate system as well as the substrate signal alone, as further described in "Methods" and Supplementary Note II. We also confirm that the origin of our extracted thin-film magnetic signal is from the terbium sublattice of the TbInO₃ film by performing X-ray magnetic circular dichroism (XMCD) at the Tb $M_{4,5}$ edge. This element-specific measurement is not susceptible to background from the substrate paramagnetism and yields data with an excellent agreement with our SQUID magnetometry measurements (Supplementary Fig. S9).

The TbInO₃ film contribution to the DC magnetic susceptibility between 1.8 K and 300 K is shown in Fig. 3a. We find that there is no ordering or spin freezing down to 1.8 K, as evidenced by the lack of splitting between field-cooled (FC) and zero-field-cooled (ZFC) curves (additional ZFC-FC curves at lower applied fields are shown in Supplementary Fig. S10). Note that the Néel temperature of possible magnetic impurities occurs from 2.4 K (Tb₂O₃) to 7.85 K (Tb₄O₇)[28] but

is not detected in our measurements. Fitting the inverse susceptibility according to the Curie−Weiss law, we extract an effective $Tb^{3+}$ moment $\mu_{eff}$ of 10 $\mu_B$, which is, within the error of our experiment, consistent with the free ion value of 9.7 $\mu_B$. We further extract a Curie−Weiss temperature $\theta_{CW} = -11$ K, which is similar to, albeit somewhat lower than, that measured in bulk crystals[9,10] and indicates frustrated antiferromagnetic interactions. By comparing the magnetic susceptibility using applied magnetic fields along the in-plane and out-of-plane directions of the film, an XY easy-plane anisotropy is identified at temperatures above 23 K, whereas the anisotropy is strongly reduced below this temperature (Supplementary Fig. S11). Such a temperature-dependent change of magnetic anisotropy was also seen in bulk TbInO₃ at a similar cross-over temperature[9,10].

Further evaluating the in-plane magnetic susceptibility down to as low as 400 mK (Fig. 3b), we find no evidence of long-range magnetic order in our films, although a deviation from the high-temperature Curie−Weiss behavior is observed below 5 K. The temperature dependence of the susceptibility in this low-temperature regime corresponds to a lowered effective magnetic moment accompanied with a smaller frustration index, and is likely related to the depopulation of exited electronic states on the terbium sites. Indeed, in bulk samples, a similar behavior is observed below 7.5 K. Based on the isothermal field dependence of magnetization at 15 and 1.8 K, we extrapolate saturation moments of ca. 7.4 and 6.7 $\mu_B$/$Tb^{3+}$ ion, showing a gradual decrease with decreasing temperature (Supplementary Fig. S12).

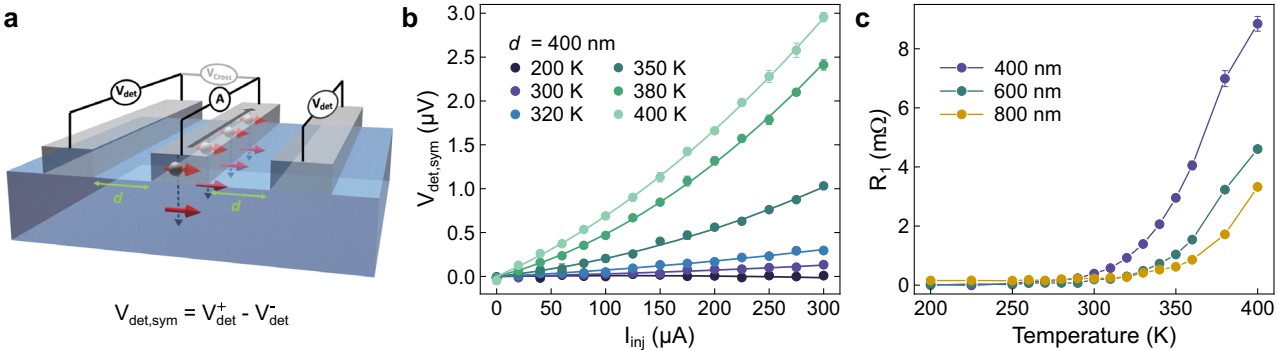

**Fig. 4 | Non-local transport signatures in TbInO₃ devices. a** Schematic of the device, consisting of parallel platinum strips on a TbInO₃ film. **b** Symmetric voltage signal measured as a function of injector current amplitude at temperatures ranging from 200 K to 400 K. Solid lines represent quadratic fits of the form $V = R_1 I + R_2 I^2$. Error bars show the standard deviation of four measurements. **c** Linear fit coefficient $R_1$ versus temperature for devices with spacing values $d$ = 400, 600, and 800 nm. Error bars represent the least squares standard deviation fit error.

Models of the depopulation of thermally populated crystal electric field levels in other rare-earth based frustrated magnetic oxides show a similar gradual reduction in saturation moment[29]. In previous studies on bulk TbInO₃, it has been suggested that the depopulation of the first excited crystal electric field (CEF) level on the Tb1 site (located at around 0.65 meV) leads to a singlet ground state with a much reduced or vanishing magnetic moment[9] and an effective magnetic lattice below 5 K with stuffed honeycomb geometry (Fig. 1c). Although our estimated saturation moment of ca. 6.5 μB/Tb³⁺ at 1.8 K (Fig. 3c) could be consistent with the stuffed honeycomb model considering a suppressed Tb1 moment[9], our data do not permit determining the exact magnetic configuration of Tb1 and Tb2 in our films. We also note that other studies of TbInO₃ bulk crystals using Raman spectroscopy[30] and inelastic neutron scattering[11], have suggested a triangular magnetic lattice with similar magnetic ground state for both Tb1 and Tb2 sites.

We next assess the AC susceptibility of the TbInO₃ film, using scanning SQUID microscopy, in the same regime below 5 K where the higher energy CEF levels are depopulated and TbInO₃ likely exists in its magnetic ground state. To characterize the substrate contributions to the thin-film susceptibility, we ion-mill etch away part of the film, exposing the bare substrate underneath (see "Methods" and Supplementary Note III for details). We use a scanning SQUID microscope in a dilution refrigerator to locally measure, with micrometer-scale spatial resolution, side-by-side the susceptibility of the bare substrate and film + substrate together. The probe field is generated by an excitation coil concentric with the SQUID's sensitive area, which applies a local field of approximately 500 μT along the c-axis of the sample. This approach therefore probes the out-of-plane susceptibility of the sample. In the weak screening limit, the susceptibility of the film is found by subtracting the bare substrate susceptibility from the susceptibility measured from the film + substrate. This allows us to determine the magnetic susceptibility specific to the TbInO₃ film down to 44 mK, an order of magnitude lower in temperature than previous work[9].

Figure 3d shows the AC susceptibility measured with the field along the c-axis of TbInO₃ from 44 mK to 4 K. At higher temperatures, the AC and DC susceptibilities exhibit similar behavior. However, in the AC susceptibility, a maximum is observed around 1 K, below which the susceptibility decreases. In addition, the temperature at which the maximum susceptibility occurs weakly depends on the applied AC frequency. The AC susceptibility appears spatially uniform within our spatial resolution.

A downturn in the AC magnetic susceptibility is observed in various magnetic systems—including spin glasses, two-dimensional spin liquid candidates, superparamagnets, systems that magnetically order

and combinations of these. In each system, different mechanisms are at play to cause a downturn in the AC susceptibility[31–35]. Our frequency dependence is reminiscent of that reported for spin freezing in spin glasses, in which disorder and geometric frustration hinder long-range order[31] (see Supplementary Note IV for an analysis of the frequency dependence). In spin glasses, the downturn occurs at a temperature at which the relaxation rates fall below the excitation frequency.

Our in-plane DC susceptibility shows paramagnetic behavior down to at least 400 mK (Fig. 3b) without a turnover as seen in the AC susceptibility. Generally, a turnover in DC susceptibility would occur at a lower temperature than in AC measurements. Extrapolating our frequency dependence using models for spin freezing transitions (see Supplementary Note IV), a turnover in DC would be expected around 600 mK. However, the DC and AC susceptibility measurements probe along different crystallographic directions in this anisotropic system raising the possibility that spin fluctuations are not freezing in the in-plane directions. In addition, the applied probe field in the DC measurements is substantially larger than the AC field, which may result in a temperature shift of a turnover in the in-plane DC susceptibility[36,37].

### Unconventional transport in Pt/TbInO₃ heterostructures

With the epitaxial stabilization of TbInO₃ thin films, we are poised to measure non-local transport in these materials. As shown in Fig. 4a, we construct devices with a measurement geometry that has been used to demonstrate magnon spin transport in insulating ferromagnets, such as yttrium iron garnet[38], and more recently antiferromagnets[39]. This setup allows for the detection of voltages generated from charge-spin interactions between the platinum layers and the TbInO₃ film, namely the spin Hall effect (injector to film) and inverse spin Hall effect (film to detector). In addition, by tracking the detector voltage readout at both positive ($V_{det}^+$) and negative ($V_{det}^-$) injection currents, the signal can be separated into a symmetric term ($V_{det, sym} = V_{det}^+ - V_{det}^-$) that excludes heat-related effects but preserves direct transport, such as magnon transport, and an asymmetric term ($V_{det, asym} = V_{det}^+ + V_{det}^-$) that includes heat-related effects, such as spin Seebeck effect[39,40]. While TbInO₃ lacks the ordered moment to form magnons, exotic carrier physics was revealed by THz conductivity in bulk TbInO₃ crystals[18]. Interestingly, this phenomena persists at temperatures well above the Curie–Weiss temperature, which traditionally defines the spin liquid regime. We thus probe the non-local transport of the TbInO₃ samples to determine whether these carriers can be manipulated electrically.

Figure 4b shows $I_{inj} - V_{det, sym}$ curves collected at temperatures between 200 K and 400 K from a device with a channel separation of 400 nm. The response grows above room temperature, indicating the apparent activation of a high-temperature transport mechanism. This

signal can have a few different contributions, each with different current dependencies: a spin Hall effect contribution, which scales linearly with $I$ ($V \propto I$), and thermal contributions due to Joule heating, which scale quadratically with $I$ ($V \propto I^2$). We perform fits of the form $V = R_1 I + R_2 I^2$ to extract coefficients corresponding to linear and quadratic components of the signal. These fits are plotted as solid lines in Fig. 4b, and the temperature dependence of the $R_1$ coefficients is shown in Fig. 4c. We see the onset of this linear scaling coefficient above 300 K, potentially indicative of high-temperature spin transport in the $TbInO_3$ film. No magnetic field dependence (in-plane or out-of-plane) was observed at any temperature using field sweeps up to 9 T (Supplementary Fig. S13). A control measurement was conducted by applying voltage across the detector and injector to measure DC current across the spacing shown in Supplementary Fig. S14. A temperature-dependent cross current was observed, with a negligibly small magnitude and a cross resistance exceeding 1 GΩ.

## Discussion

In summary, we have demonstrated the epitaxial thin-film synthesis of a quantum spin liquid candidate: magnetically frustrated $TbInO_3$. Our films are single-phase and highly crystalline with a low surface roughness. They exhibit high-temperature improper ferroelectricity that emerges already during synthesis and displays a nanoscale domain pattern with an estimated local polarization of approximately $7 \, \mu C \cdot cm^{-2}$. We explored the manifestation of spin liquid behavior in these films, where advanced magnetometry points to an absence of long-range ordering down to at least 0.4 K despite a Curie–Weiss temperature of −11 K. At temperatures below 5 K, we see a change in the in-plane magnetic susceptibility indicative of $Tb^{3+}$ magnetic moments entering their ground state and the possible development of a stuffed honeycomb magnetic lattice. An anomalous downturn in the AC magnetic susceptibility along the out-of-plane $c$-axis is found to occur in our films at temperatures below 1 K, indicating a slowing down of spin fluctuations. Although this behavior does not necessarily contradict the presence of a quantum spin liquid ground state, spin freezing or related disorder effects cannot be excluded. Strikingly, enabled by our thin film samples, nonlocal transport measurements indicate that unconventional carrier transport unrelated to magnetic long-range order occurs in this system at temperatures well above $|\theta_{CW}|$ that, to the best of our knowledge, has not been observed previously. In light of the recent report of exotic room-temperature carrier dynamics in bulk crystals of $TbInO_3$ probed by THz conductivity[18], our observation is further testament to the richness of exotic physics found in this system beyond the low-temperature regime of phenomena in quantum spin liquids. Future work could explore whether this exotic transport is unique to the spin liquid candidate $TbInO_3$ or more generic to other highly entangled magnets.

## Methods
### Thin-film growth
The hexagonal $TbInO_3$ films were grown on (111) $(ZrO_2)_{0.905}(Y_2O_3)_{0.095}$ (or 9.5 mol% yttria-stabilized zirconia) substrates, denoted YSZ, by reactive-oxide MBE using a Riber Compact21 chamber. The substrates were kept at a temperature of approximately 860 °C during thin-film growth as measured by an optical pyrometer focused on a platinum backside coating. The base pressure before growth was better than $10^{-9}$ torr. The deposition was performed under distilled (up to 90% $O_3/O_2$) ozone from Heeg Vacuum Engineering with a partial pressure of $5 \times 10^{-7}$ torr to $1 \times 10^{-6}$ torr. Indium and terbium were evaporated from elemental sources; the flux was roughly calibrated using a quartz crystal microbalance. The growth window was then optimized through monitoring of the RHEED patterns during deposition. RHEED images are shown in grayscale or black/green/white colorscale, where black and white correspond to the minimum and maximum intensity, respectively. Phase-pure $TbInO_3$ only forms under a very narrow range

of composition—indium-rich films were identified from 3-D islands of $In_2O_3$ apparent both on the RHEED images and in post-synthesis atomic-force microscopy (AFM) images (Supplementary Figs. S1 and S2). Terbium-rich films were identified by the presence of a Tb-O phase in the RHEED images.

### Structural characterization
The film structure was characterized by X-ray diffraction (XRD) using a four-circle Malvern Panalytical Empyrean diffractometer equipped with a Ge(220) × 2 monochromator on the incident side. The film surface topography was characterized with a MFP-3D Origin+ Asylum AFM.

X-ray Absorption Spectroscopy (XAS) and XMCD at the Tb $M_{4,5}$ (1240–1280 eV) edge as a function of temperature (20–100 K) and magnetic field (0–3.5 T) were performed at beamline 4.0.2 at the Advanced Light Source, Lawrence Berkeley National Laboratory, using a superconducting vector magnet. The measurements were conducted in normal incidence geometry with the magnetic field parallel to the beam and recorded by measuring the total electron yield. Each spectrum shown is an average of 16 individual scans.

### Scanning transmission electron microscopy
Cross-sectional scanning transmission electron microscopy (STEM) specimens were prepared using an FEI Helios 660 Focused Ion Beam with a final milling step at a beam energy of 2 keV to reduce surface damage. High-angle annular dark-field STEM measurements were performed either on a JEOL ARM 200F or a Thermo-Fisher Scientific Titan Themis Z G3, both operating at 200 keV beam energy. The convergence angle was either 19.6 or 22 mrad and the collection angle range was ~68 mrad–280 mrad. STEM micrographs and FFT data are presented with a linear grayscale colormap where white corresponds to the maximum intensity. Local strain variations were mapped using the phase lock-in technique developed in Ref. 41. For mapping the improper ferroelectric domains and trimerization, atomic column positions were located by 2D Gaussian fitting using the Python package Atomap[42]. Trimerization amplitude was defined as the difference in the average [001] displacement of the "up" and "down" terbium atoms. The average displacement amplitude in the film was calculated from averaging over $N \sim 22500$ atomic positions. The phase of polarization domains was mapped based on fitting to a sinusoid using the method detailed by Holtz et al.[23].

### Magnetic measurements
The bulk magnetic properties were determined using a Quantum Design MPMS 3 superconducting quantum interference device (SQUID) magnetometer in DC scan mode. The samples were mounted with GE Varnish on a quartz rod for measurements with the applied magnetic field in the plane of the film. For measurements with the field applied out of plane, the samples were mounted inside a plastic straw. Before all measurements, the samples were backside polished to remove the thermal Pt coating used during deposition. To subtract the highly variable magnetic background signal of the YSZ substrates, the film side of the samples was polished off after magnetic characterization, and the bare substrate was measured again under the same conditions as the thin-film sample[43]. The substrate signal was then subtracted point by point from the signal of the thin-film measurement (for further details, see Supplementary Note II).

Low-temperature magnetic measurements were recorded using a Quantum Design MPMS 5S magnetometer with the iQuantum He3 attachment, allowing for measurements below 0.5 K. Installation of the He3 cooling system was conducted in accordance with instructions from Quantum Design. ZFC and field cooled susceptibility measurements were conducted at a field of 2000 Oe from 0.4 to 300 K.

Scanning SQUID measurements were performed using a SQUID susceptometer to locally probe the susceptibility of the substrate and

the TbInO$_3$ film. The SQUIDs consist of an inner pick-up coil (inner radius = 0.75 μm, outer radius = 1.6 μm) that is coupled to the SQUID circuit used to readout magnetic flux, and an outer field-coil (inner radius = 3 μm, outer radius = 6 μm) through which we flow an AC current and apply a local field to the sample. We measure the susceptibility of the bare substrate and the film and substrate as a function of temperature and frequency of the AC current flowing through the field-coil. The susceptibility of the TbInO$_3$ is found by subtracting the signal measured on the bare substrate from that measured on the film and substrate. See Supplementary Note III for details.

**Non-local spin transport**

The non-local devices were patterned directly on the TbInO$_3$ film. The metal strips with various spacing $d$ and a channel length of 100 μm and width of 350 nm were patterned by e-beam lithography (JEOL JBX-6300FS) using the PMMA 950 resist with a lift-off process, exposed with a dose of 850 μC·cm$^{-2}$. Next, the sample was treated with oxygen plasma (100 W, 10 s) to ensure a clean interface. Finally, 15 nm of platinum was deposited by an e-beam evaporator with a base pressure of $2 \times 10^{-6}$ torr, equipped with a quartz crystal monitor for thickness feedback. The contact pads were subsequently patterned by a standard SPR-220 based process using a lift-off technique with the metals (300 nm Au/10 nm Ti) deposited by an e-beam evaporator.

The non-local measurements were performed in a PPMS system (DynaCool, Quantum Design) using a Keithley nanovoltmeter 2182A and a Keithley 2450 source-meter-unit. The measurements were done by sourcing current up to ± 300 μA to the injector and recording the voltage from the detector, with the reversing current technique to track the detector voltage at both injection current directions ($V^+$, $V^-$) where the symmetric ($V_{sym} = V^+ - V^-$) and asymmetric ($V_{asym} = V^+ + V^-$) components can be separated.

**Computational methods**

We perform DFT calculations using the Vienna ab initio Simulation Package[44]. We employ a 600 eV plane-wave cutoff, a $4 \times 4 \times 4$ $k$-point mesh in a 30-atom computational cell, a $1 \times 10^{-7}$ eV energy convergence criterion, and a 2 meV Å$^{-1}$ force convergence criterion for structural relaxations. We use the Perdew-Burke-Ernzerhof functional[45] and the following projector-augmented wave pseudopotentials[46]: Tb_3 ($6s^2 5p^6 5d^1$), In_d ($4d^{10} 5s^2 5p^1$), and O ($2s^2 2p^4$). We use density functional perturbation theory to compute Born effective charges, which we combine with polar displacement amplitudes to obtain the electrical polarization. For comparison, we also perform DFT+$U$ + spin-orbit coupling (SOC) calculations, which treat the Tb $f$ electrons explicitly. For these calculations, we use the Tb ($4f^8 5s^2 6s^2 5p^6 5d^1$) pseudopotential and make use of the Lichtenstein formulation[47] of DFT + $U$ with $U = 8$ eV and $J = 1$ eV. Our $U$ choice is consistent with prior DFT work on TbInO$_3$[13]. We impose a ferromagnetic arrangement of the Tb spins, and incorporate SOC self-consistently in all calculations. All other computational parameters are the same as those given above. Computed lattice parameters and bandgaps show good agreement with their experimental values. For more details, see Supplementary Note I.

## Data availability

Source data are provided in the Source Data file. Additional data which support the findings of this study are available from the corresponding authors upon request. Source data are provided with this paper.

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

## Acknowledgements

The authors would like to thank Sandesh Kalantre, David Goldhaber-Gordon, EliseAnne Koskelo, Megan Holtz, and Jennifer Hoffman for fruitful discussions. We acknowledge support on the XMCD measurements from Christoph Klewe at the Advanced Light Source. This work was supported by the Air Force Research Laboratory, Project Grant FA95502110429. This work used resources of the Advanced Light Source, which is a DOE Office of Science User Facility under contract no. DE-AC02-05CH11231. Electron microscopy was carried out through the use of MIT.nano facilities at the Massachusetts Institute of Technology. Additional electron microscopy work was performed at Harvard University's Center for Nanoscale Systems, a member of the National Nanotechnology Coordinated Infrastructure Network, supported by the NSF under Grant No. 2025158. Low-temperature SQUID measurements were performed at the Laukien-Purcell Instrumentation Center with assistance from Claire Casaday, Kevin Anderton, and Dongtao Cui. Nanofabrication work was performed at the University of Michigan Lurie Nanofabrication Facility. J.A.M. acknowledges support from the Packard Foundation and Gordon and Betty Moore Foundation's EPiQS Initiative, Grant GBMF6760. J.N. acknowledges support from the Swiss National Science Foundation under Project No. P2EZP2_195686. C.T. acknowledges support from the Swiss National Science Foundation under Project No. P2EZP2_191801. H.E.S. and I.E.B. acknowledge support from the Rowland Institution at Harvard.

## Author contributions

Thin films were synthesized by J.N., M.A.A., C.M.B. and J.A.M. The films were characterized by spin transport by T.C., S.D. and J.T.H., by scanning SQUID by A.K. and K.C.N., by SQUID by J.N., M.A.A. and C.T., by XAS by E.M. and A.D.L.T. and by TEM by H.S., M.A.A. and I.E.B. DFT calculations were performed by N.P., K.T. and E.A.N. and theoretical models were constructed by J.H.S. and E.A.K. The paper was written by J.N. with assistance from M.A.A. and J.A.M. and with input from all of the authors. J.N. and J.A.M. conceived of and guided the study.

## Competing interests

The authors declare no competing interests.
