## [Transparent Peer Review file · Nature Communications]

Signatures of Quantum Spin Liquid State and Unconventional Transport in Thin Film TbInO_3

Corresponding Author: Dr Johanna Nordlander

Version 0:

Reviewer comments:

Reviewer #1

(Remarks to the Author)

In the ultrathin stoichiometric TbInO_3 grown on (111)-oriented YSZ substrates, the authors reported the frustrated magnetism similar to that in bulk TbInO_3 , improper ferroelectricity, and unconventional carrier transport. The thin-film realization of frustrated magnetism is important for developing its expected electronic applications. Although there are still many issues regarding understanding ferroelectricity and strain in the TbInO_3 thin films, the most important of frustrated magnetism is convincing. Hence, I agree with the publication after significant revisions.

1. Strain condition

In Fig S3-S4, the RSM and STEM strain map give the strain condition of TbInO_3 thin film. However, these results have not got sufficient discussion in the text. The strain is significant for influencing material structure and achieving functionality in epitaxial thin film.

The authors described that "In-plane x-ray reciprocal space mapping and strain mapping in STEM reveal that the films are partially relaxed with respect to the substrate lattice above a thickness of 8 nm". Does the retention of epitaxial strain play a role in the frustrated magnetism and ferroelectricity of thin films? In addition, if the authors want to illustrate the relationship between thickness and strain through STEM data, they cannot provide just a STEM image of one sample to support this conclusion.

Fig S4 is not clear enough. In the STEM image, the interface of YSZ/ TbInO_3 looks not sharp. Is there contains ~3 nm thickness region of chemical diffusion?

The authors found that "increased ϵ_{xx} at the interface" (Figure S4). Near the substrate, there is a special horizontal ~ +1.5% strain region of 2 nm thickness between compressive-strain regions. In addition, the in-plane strain of TbInO_3 thin film exhibits significant local non-uniformity. What are the sources of these phenomena? Does the substrate strain play a role in achieving frustrated magnetism and improper ferroelectricity?

2. Ferroelectric domains

The authors showed the improper ferroelectricity of TbInO_3 thin films through a large amount of polar mapping results. However, I think the organizations of the data and text in this section have some issues making me concern. Fig2 and Fig S6-S8 provide so many similar polar mapping results. Are they from the same sample? What does the author want to illustrate by providing so many images? On the contrary, the mechanism of improper ferroelectricity is still unclear.

Figure 2 shows the up/down polar domains in TbInO_3 thin films, which differ from the previously observed single domain states. Can we identify these domains on a relatively large scale, such as using Dark Field-TEM, or corresponding strain map, rather than just polar maps? More importantly, the origin of these nanoscale domains should be discussed.

In Fig S5a, many local contrast differences can be found in the HAADF-STEM of TbInO_3 thin films. Could we assume it comes from chemical variations (Z contrast)? Are they related to the emergence of ferroelectric domain structures?

Compared to the rough discussions of strain and ferroelectricity (the authors put all the obtained data in the article), the study

of frustrated magnetism is very professional. They clearly showed the frustrated magnetism of TbInO₃ thin films while removing the influence of the substrate.

Reviewer #2

(Remarks to the Author)

In this manuscript, the authors report on the fabrication of epitaxial TbInO₃ thin films and their investigation of magnetization and non-local transport properties. Quantum spin liquids (QSLs), with their highly quantum entangled states, have generated considerable interest due to potential applications in fault-tolerant quantum computation. The fabrication of thin films of the QSL materials represents a crucial first step toward potential device applications. TbInO₃ has recently emerged as a QSL candidate material, exhibiting ferroelectricity and exotic carriers at high temperatures. However, its ground state remains elusive, particularly regarding the magnetic state of Tb1 sites.

While the authors' conventional SQUID magnetization measurements show results similar to bulk crystals (albeit with some temperature scale discrepancies), authors have also employed previously unexplored experimental probes, including scanning SQUID and non-local transport experiments. Given the limited reports on the fabrication of the QSL thin film, this research direction is significant, and the TbInO₃ film growth appears to be successful. However, there are several issues regarding the analysis and discussion of the experimental results that the authors need to address. More rigorous analysis and discussion are needed to evaluate whether the paper is worthy of publication in Nature Communications.

1. The authors mention that the magnetic susceptibility measured by conventional SQUID magnetometry includes a contribution from the impurity within the substrate at low temperatures. The relative magnitudes of the film+substrate versus substrate-only signals need to be clarified. If the substrate signal substantially exceeds the film signal, the uncertainty in the subtracted result could be significant. Including the raw data from both measurements would enable readers to assess the uncertainty of the results. Furthermore, the methodology for subtracting the substrate background signal should be clarified. While the authors mention the "point-by-point" subtraction, they should specify whether this refers to subtraction of SQUID scan curves or subtraction of fitted magnetization values. The former approach would be preferable, but the current methodology is not adequately described.

2. The temperature dependence of magnetic susceptibility in the high-temperature region suggest the Tb³⁺ ion moment is 10 μ B. The authors attribute the deviation below 5K to reduced occupation of CEF excited states at the Tb1 site. This interpretation is consistent with the reduced moment of 6.5 μ B derived from the field dependence of magnetization at 1.8K. Based on this interpretation, it would be important to estimate the saturation magnetic moment of Tb³⁺ ions from the field dependence of magnetization above 5K, where one would expect values closer to 10 μ B. Given the ongoing debate regarding the magnetic state at the Tb1 site, clarifying this point would be essential.

3. The authors present AC susceptibility measurements at dilution refrigerator temperatures using scanning SQUID that requires impressive technical capability. However, the discussion of these sophisticated experimental results requires substantial elaboration. The current manuscript does not adequately explain the implications of the observed decrease in the AC susceptibility below 1K. Several critical questions remain unaddressed. What types of QSL states would be consistent with these measurements? What mechanisms could explain the discrepancy between AC and DC susceptibility temperature dependencies below 1K? Is it appropriate to attribute the phenomena observed exclusively in AC susceptibility to dynamic effects? I would like to encourage the authors to include a more comprehensive interpretation of these experimental findings.

4. Following from the previous point, the authors suggest that the disorder effects cannot be ruled out as the origin of the decrease of the AC susceptibility, but this requires further clarification. What specific possibilities are they considering? Given that scanning probe measurements were employed, couldn't spatial dependence data be used to evaluate the potential disorder effects?

5. The authors attempt to correlate their non-local transport signals with exotic carriers previously observed in the THz measurements. However, there is a significant discrepancy between these observations: while THz measurements reported temperature-independent exotic carriers between 150K-300K, the present study shows that the R1 signals related to SHE emerge only above 300K. These two observations cannot be straightforwardly reconciled, suggesting a gap in the current discussion. A more thorough discussion is warranted on the origin of the observed non-local transport. In particular, the observation of magnetism-related transport phenomena at temperatures more than an order of magnitude above the Curie-Weiss temperature challenges conventional understanding. While a complete explanation may be challenging, the authors should explore several possible scenarios to address this unusual behavior.

6. Regarding the non-local transport data, the authors present only the symmetric component. What is the behavior of the raw data? From the current presentation, it is impossible to assess whether the symmetric part attributed to SHE is sufficiently large enough to be accurately measured compared to the raw data. The authors should provide a more comprehensive presentation of their transport measurements to substantiate their conclusions.

Overall, while the successful fabrication of the epitaxial thin films of the QSL candidate material represents a significant achievement and the thin film preparation enables novel measurements such as non-local transport, the discussion and analysis of these results are insufficient. I cannot recommend the manuscript for publication in Nature Communications in its current form. If the authors wish to emphasize the importance of thin film fabrication, they need to provide more elaborate discussions and comprehensive analyses of the experimental results that have become accessible through thin film

preparation.

Reviewer #3

(Remarks to the Author)

This paper by Nordlander et al. demonstrates the growth of epitaxial thin films of TbInO_3 on YSZ substrates. Authors have put exceptionally great efforts into optimizing the growth conditions to produce phase pure films of TbInO_3 , as confirmed via in-situ RHEED, XRD, AFM, and STEM results.

The presence of ferroelectric domains in the films has been shown on multiple samples, and the results are in close agreement with the bulk counterparts.

The absence of magnetic order down to at least 0.4 K (despite a Curie-Weiss temperature of -11K) is inferred using AC and DC -susceptibility (via scanning SQUID) along different crystalline orientations and XMCD measurements, which gives signatures of promising quantum spin liquid state in these films.

Additionally, unconventional transport in these films using Pt/TbInO_3 heterostructures is also demonstrated, indicating the presence of unusual charge dynamics in this system near and above room temperatures (well above Curie-Weiss temperature). Such observations make this system an exciting candidate for exploring physical phenomena beyond the low-temperature regime.

On top of all, these results were backed up with theoretical calculations performed using DFT + U + SOC along with DFPT calculation.

The paper is well-written and explores interesting areas of physics, combining thin film magnetism (quantum spin liquids in particular, which is very uncommon and sparingly reported), ferroelectricity, and non-local transport, which meets the interest of the broad audience of Nature Communication journal.

I have the following comments/questions for the author, which need explanation.

1. The title "Signatures of Quantum Spin Liquid Ground State in Epitaxial Thin Films of TbInO_3 " seems very specific, demonstrating a particular property, which, in my view, can be made general to involve the other results discussed in the manuscript. (suggestion).
2. The ferroelectric domain (using STEM) of many different films has been shown in the main text and supplemental information (SI). I found that all of these are nearly 15 nm in thickness. It would be interesting to know how these domains evolve with the thickness of the film.
3. The authors demonstrated the magnetic properties of TbInO_3 film with a thickness of 38 nm in Fig 3. These films appear relaxed according to the RSM data in Fig. S3 (SI). I'm wondering why there is an inconsistency in the thickness of the films between the two measurements (magnetic and ferroelectric). I would like to see the magnetization results for the thinner films (in a strained state) and the relaxed films to check how magnetism evolves with confinement to 2D.
4. The in-plane magnetic susceptibility (χ_{ab}) shown in Fig 3(a) has significantly higher values (even at 300K) compared to other materials in the paramagnetic phase ($\sim 10^{-3}$ to 10^{-4} emu /Oe /mol). What can be the origin of such high moments at room temperature?
5. Also, in Fig 3(a), a tiny variation is observed in ZFC and FC curves in the temperature range of 100-250K. Can authors comment on whether it comes from the film or has another origin?
6. The methods section indicates that the ZFC-FC measurements were conducted in a field in 2000 Oe. Is it possible to check the response of the films in a lower-applied field, as it can better probe the weak spin interactions in the system?
7. Electron hopping conduction has proved to be a simple way of verifying the dimensionality of electron interaction present in the system. Have the authors tried fitting the resistivity of the films using such formalism (if it applies to these films)?
8. The presence of non-local transport in the heterostructures of these systems is remarkable and needs further exploration. I request the authors to emphasize more on its significance and possible use in these systems. This will strengthen the overall manuscript.
9. What are the author's thoughts about the effect of dimensional confinement (in thin film geometry) on the TbInO_3 system? Are there changes in properties that distinguish them from their bulk equivalents?

Version 1:

Reviewer comments:

Reviewer #1

(Remarks to the Author)

This work's main contribution is demonstrating the thin-film manifestation of exotic properties, such as a potential quantum spin liquid ground state in the first TbInO_3 thin film. The authors clearly show this point with detailed data. In my opinion, the article can be published after some minor modifications. I only have some questions and suggestions about the details.

1. The presented data combine results from films of varying thicknesses, such as the ferroelectric domain pattern (Fig. 2, 15 nm), SQUID magnetometry (Fig. 3, 38 nm), and isothermal magnetization curves (Fig. S13, 29 nm). Why were different thicknesses selected for those measurements?

2. The authors have shown that films with varying thicknesses exhibit different misfit strains (Figs. S3-S5). However, they did not address the implications of these variations. Although the films with different thicknesses display similar ferroelectric

domain structures, other possible differences—such as magnetic properties—were not discussed.

3. The authors mention a cubic transitional layer in TbInO₃ films. Figs. S4/S5 appear compressed during editing, making identifying the transitional layer from STEM images difficult. Based on the strain maps, 8 nm film has a cubic transitional layer (strain $\gamma_x \approx 0$), but the transitional region in 15 nm film appears non-cubic. Additionally, I suggest adding a brief discussion about the transition layer.

4. Please provide the technical details of strain mapping in Methods.

5. Line 339. "...using an FEI Helios 660 Focused Ion Beam (FIB) with a final milling step of 2 keV to reduce surface damage." I think that 2 keV should be 2 kV.

Reviewer #2

(Remarks to the Author)

In the revised manuscript, the authors have expanded the discussion of their experimental results. Although some aspects remain unclear, it is often challenging to fully resolve all issues in strongly correlated systems, particularly in quantum spin liquids (QSLs). Given the significance of the subject and the scarcity of reports on the fabrication of QSLs, I believe that the current manuscript meets the standards for publication in Nature Communications.

Reviewer #3

(Remarks to the Author)

On careful examination of the revised manuscript and rebuttal, I found that the authors have worked extremely well in answering the comments, thoroughly modifying the manuscript and adding useful information to support the findings. These satisfactorily address my comments and concerns, and I hence recommend it for publication.

**Review Response for Signatures of Quantum Spin Liquid State
and Unconventional Transport in Thin Film TbInO₃**

Johanna Nordlander,* Margaret A. Anderson,* Tony Chiang, Austin Kaczmarek, Nabaraj Pokhrel, Kuntal Talit, Spencer Doyle, Edward Mercer, Christian Tzschaschel, Jun-Ho Son, Hesham El-Sherif, Charles M. Brooks, Eun-Ah Kim, Alberto de la Torre, Ismail El Baggari, Elizabeth A. Nowadnick, Katja C. Nowack, John T. Heron, and Julia A. Mundy

REVIEWER 1

In the ultrathin stoichiometric TbInO₃ grown on (111)-oriented YSZ substrates, the authors reported the frustrated magnetism similar to that in bulk TbInO₃, improper ferroelectricity, and unconventional carrier transport. The thin-film realization of frustrated magnetism is important for developing its expected electronic applications. Although there are still many issues regarding understanding ferroelectricity and strain in the TbInO₃ thin films, the most important of frustrated magnetism is convincing. Hence, I agree with the publication after significant revisions.

We thank the reviewer for their thoughtful review and suggestions to improve our manuscript.

1. Strain condition

In Fig S3-S4, the RSM and STEM strain map give the strain condition of TbInO₃ thin film. However, these results have not got sufficient discussion in the text. The strain is significant for influencing material structure and achieving functionality in epitaxial thin film.

We agree with the reviewer that strain can have an important influence on thin film properties and functionality. We expanded our discussion of strain in the text as presented in response to the reviewer's following point. To better explore the potential impact of film thickness and strain on ferroelectric and magnetic properties, we have conducted SQUID magnetometry and STEM imaging on the 8 nm film presented with a fully-strained RSM in Fig. S3c. With this new data we are able to compare the behavior of this thinner film with that of the thicker films and better examine the effects of film thickness and epitaxial strain.

The authors described that "In-plane x-ray reciprocal space mapping and strain mapping in STEM reveal that the films are partially relaxed with respect

* Equal contribution

to the substrate lattice above a thickness of 8 nm". Does the retention of epitaxial strain play a role in the frustrated magnetism and ferroelectricity of thin films? In addition, if the authors want to illustrate the relationship between thickness and strain through STEM data, they cannot provide just a STEM image of one sample to support this conclusion.

While our primary evidence for partial relaxation comes from x-ray reciprocal space mapping, we have performed STEM imaging and strain mapping on the 8 nm film to better demonstrate the relaxation above 8 nm as suggested by the reviewer. We have expanded the discussion of our STEM strain mapping results and their agreement with diffraction data in the text as follows:

Figure 1(e) shows a $\theta - 2\theta$ XRD scan which confirms the purely (001)-oriented layered hexagonal structure. Laue oscillations around the (004) reflection of TbInO_3 are testament to sharp film interfaces, as further confirmed by a low surface roughness determined by atomic-force microscopy (Fig. S1). The in-plane lattice mismatch between bulk $\text{TbInO}_3(0001)$ and a 30° rotation of $\text{YSZ}(111)$ is -0.8%. In-plane x-ray reciprocal space mapping (RSM) reveals that the films gradually relax with respect to the substrate lattice with increasing thickness (Fig. S3). While at a thickness of 8 nm, the film lattice is fully aligned with the substrate, the 15 nm and 29 nm films show gradual relaxation. Likewise, local strain maps based on STEM images show that the 8 nm film has a uniform ϵ_{xx} across the interface (Fig. S4). In the 15 nm film, except for a region of increased ϵ_{xx} near the film-substrate interface, there is a slight negative ϵ_{xx} within the film (Fig. S5), which reveals partial relaxation of the in-plane strain, as expected from the x-ray RSM data.

We updated the existing strain mapping supplemental figure and added a new one corresponding to the 8 nm film (Figs. R1-R2, S4-S5 in the supplement).

We have added additional discussion of the impact of strain on film polarization and magnetic properties as well:

The TbInO_3 thin films display all six trimerization domains (Fig. 2(b-d), Figs. S7-8) with a smaller domain size and thus larger number of domain walls in

FIG. R1. Strain mapping in an 8 nm TbInO₃ on YSZ(111). (a) A HAADF-STEM micrograph of the 8 nm thick TbInO₃ film (top) and YSZ substrate (bottom) and a transitional layer with a cubic structure in between. (b-c) Strain maps of ϵ_{xx} and ϵ_{yx} in the region shown in (a) derived from a lock-in phase analysis. Uniform ϵ_{xx} across the interface (b) suggests the film is fully strained to the substrate. Finite ϵ_{yx} clearly identifies regions with the ideal hexagonal TbInO₃ structure.

FIG. R2. Strain mapping in 15 nm TbInO₃ on YSZ(111). (a) A HAADF-STEM micrograph of the interface between the TbInO₃ film (top) and YSZ(111) substrate (bottom) with a cubic transitional layer at the interface. (b-c) Strain maps derived from a lock-in phase analysis showing increased ϵ_{xx} at the interface and small negative ϵ_{xx} within the film (b), which indicates partial relaxation of the film, and homogeneous, finite ϵ_{yx} within the film (c).

comparison to the bulk crystals of TbInO₃ [10]. A thinner 8 nm TbInO₃ also shows a similar domain pattern (Supp. Fig. S7).

and:

The thinner sample shows a slightly smaller displacement (Supp. Fig. S8),

possibly due to strain or interface clamping imparted from the substrate. While the displacement in the thicker film corresponds to a polarization similar to the reported value of the ferroelectric polarization of the hexagonal manganites and ferrites, it is larger than the electrical polarization measured in bulk TbInO_3 [10] at 77 K, despite our slightly smaller distortion.

Although we also acquired magnetization data from the 8 nm TbInO_3 film, we emphasize that this is an incredibly challenging measurement which relies especially heavily on the careful background removal since the relative contribution from the substrate signal dominates over that stemming from the much reduced volume of the ultrathin TbInO_3 film (see Fig. R3a). We also note that the presence of a 2-nm thick transitional layer of a cubic phase at the substrate interface (as seen in Supp. Fig. S4a (R1a)) constitutes a significant portion (about 25%) of the total film thickness. Combined, these added uncertainties in the ultrathin regime of TbInO_3 render a quantitative analysis of the magnetic susceptibility unreliable. Having noted this important caveat, we show the zero-field cooled (ZFC) and field cooled (FC) curves of the magnetic susceptibility for the 8-nm sample in Fig. R3b. The data show no signatures of phase transitions or spin freezing down to at least 1.8 K. However, the mixed structure of the film at this thickness prevents further conclusions about the TbInO_3 -specific magnetic ground state.

Fig S4 is not clear enough. In the STEM image, the interface of YSZ/TbInO3 looks not sharp. Is there contains ~3 nm thickness region of chemical diffusion?

The reviewer has a keen eye in noting the presence of a 2-3 nm thick transitional interface layer which does not exhibit the expected hexagonal ABO_3 structure. The cubic, terbium-rich layer is formed as the film transitions from the cubic structure of the YSZ substrate to the bulk-stable hexagonal layered structure of the TbInO_3 film. We note that this layer results from a restructuring of the film and not from chemical diffusion of the YSZ substrate. The hexagonal portion of the film is highlighted in the ϵ_{yx} map. The captions of the updated Figs. S4-5 are modified to highlight the transitional layer in both micrographs.

FIG. R3. Temperature-dependent magnetization from SQUID magnetometry on an 8-nm sample grown on YSZ(111). The data was acquired with an applied magnetic field of 2000 Oe along the ab plane. (a) The induced magnetic moment under zero-field cooling (ZFC) and field-cooling (FC) conditions measured on the thin-film sample and the corresponding substrate measurement under the same conditions after removing the film. (b) The film-only signal is extracted by normalizing the sample data in (a) by the substrate measurement as described in Methods. Paramagnetic behavior is observed in the 8-nm film down to 1.8 K without splitting between the FC and ZFC curves.

The authors found that “increased ϵ_{xx} at the interface” (Figure S4). Near the substrate, there is a special horizontal +1.5% strain region of 2 nm thickness between compressive-strain regions. In addition, the in-plane strain of TbInO₃ thin film exhibits significant local non-uniformity. What are the sources of these phenomena? Does the substrate strain play a role in achieving frustrated magnetism and improper ferroelectricity?

The increased strain region at the interface corresponds to the cubic transitional layer discussed in the response to the reviewer’s previous comment. The cubic structure is stabilized by the cubic substrate before the film adopts the bulk-stable hexagonal ABO_3 structure. Within the 15 nm film, the partial relaxation of the strain appears to occur at the interface between the ideal TbInO₃ film and the transitional layer. Present in every film, the layer is not known to influence the magnetic or ferroelectric properties of the film beyond the

impacts of imparting epitaxial strain.

The ‘significant local non-uniformity’ noted by the reviewer is an artifact of the strain mapping technique which commonly shows some local variation even where the strain is constant (see, for example, non-uniformity in the substrate reference region). This noise is especially pronounced in regions where the strain is close to zero (consider the apparent difference in uniformity within the film in R1b compared to R1c, for example).

As noted above, both improper ferroelectricity and frustrated magnetism is present in (unstrained) bulk TbInO_3 crystals. Hence, the substrate strain is not the source of these properties, but may influence their manifestation in the strained thickness regime.

2. Ferroelectric domains

The authors showed the improper ferroelectricity of TbInO3 thin films through a large amount of polar mapping results. However, I think the organizations of the data and text in this section have some issues making me concern. Fig2 and Fig S6-S8 provide so many similar polar mapping results. Are they from the same sample? What does the author want to illustrate by providing so many images? On the contrary, the mechanism of improper ferroelectricity is still unclear.

As the reviewer points out, Figs S6-S8 contained information which overlapped with that of Fig 2 in the main text. The additional data sets in the supplement were intended to demonstrate that the region mapped in Fig 2 was a typical region of the film with similar domain structure seen in other regions (S6-S8) and further provide additional context including statistics of the polarization in each region and visualizations of the mapping within an extended field of view including the substrate-film interface and film surface. To reduce this redundancy, we have removed Figs S6-8 and instead adjusted Fig. 2 based on updates to the polarization mapping code and added Figs. S7 and S8 (R4 and R5) which show new HAADF-STEM and polarization mapping within the 8 nm film and collated statistics which combine several unique regions of each film, respectively.

The main focus of our manuscript is to demonstrate the thin-film manifestation of exotic properties such as a potential quantum spin liquid ground state in the first TbInO_3

FIG. R4. Improper ferroelectric domain mapping in an 8 nm TbInO₃ epitaxial thin film. (a) A map of the local distortion amplitude overlaid on a HAADF-STEM micrograph. (b) A map of the improper ferroelectric domains in the same region as (a).

thin films. Because our work reports the very first thin-film synthesis of this compound, we provide an extensive structural characterization of our films, including mapping to the structural distortion that drives the ferroelectric state reported in bulk TbInO₃ [1] which is further supported by our DFT calculations. A detailed discussion of the mechanism of improper ferroelectricity in this material is, however, beyond the scope of the current study. It has already been shown that in isostructural improper ferroelectrics such as YMnO₃, the mechanism driving the improper ferroelectricity in bulk, which is well-understood [1–5] (listed as 10,14-17 in the main text), is retained in thin films down to the ultrathin regime [6]. We add these additional references to the introduction as follows:

The material undergoes a structural transition at high temperature: a coordinated tilting of the InO₅ trigonal bipyramids and an accompanying distortion of the terbium layers breaks inversion symmetry to generate an improper ferroelectric [10,14-17].

Figure 2 shows the up/down polar domains in TbInO₃ thin films, which differ from the previously observed single domain states. Can we identify these domains on a relatively large scale, such as using Dark Field-TEM, or

FIG. R5. Statistics of improper ferroelectric domain mapping in thin films of TbInO_3 . (a) A histogram of local distortion magnitudes extracted from 10 unique regions of an 8 nm film with a mean of 33.06 ± 0.09 pm ($N = 8810$). (b) The distribution of the magnitude and phase of the polarization of the regions considered in (a). (c) A histogram of local distortion magnitudes extracted from 7 unique regions of a 15 nm film with a mean of 34.21 ± 0.05 pm ($N = 26518$). (d) The distribution of the magnitude and phase of the polarization of the regions considered in (c).

corresponding strain map, rather than just polar maps? More importantly, the origin of these nanoscale domains should be discussed.

The reviewer makes an excellent suggestion to use dark-field TEM to attempt to identify polar domains on a larger scale. Dark-field TEM could potentially be used to map the polar domains in our samples at a larger scale. However, the domain contrast in thin film

samples is known to be weaker than in bulk crystals, likely due to the smaller domain size [7] and polarization magnitude. Furthermore, while dark-field TEM can differentiate up versus down polarization, it cannot uniquely identify each of the six distinct polarization domains (presented in Fig. 2c) [1, 7, 8]. Thus, mapping the atomic distortion with STEM provides richer information about polar domain structure [9].

Using a strain map to try to identify polarization domains is also an intriguing idea. We have attempted to use the phase lock-in strain mapping technique, analogous to geometric phase analysis, to more easily map the domain structure, but we saw no correlation between the phase or strain maps and the local polarization domains (Fig. R6). This technique can be applied to any peaks in the fourier transform, so we tested a variety of superlattice peaks as well as the film peaks used to map the strain in S4-S5. The null result is not surprising because the polar domains in TbInO_3 are not ferroelastic.

Because the domains are small scale in the thin films, mapping based on atomic position has the ideal resolution to map the fine domain structure in greater detail than achievable with larger-scale techniques while still mapping over several domains in one field of view and giving a sense of the long-range structure. Furthermore, mapping the atomic polar distortions is directly interpretable whereas in techniques such as dark-field TEM, there can be ambiguity surrounding the origin of contrast. As stated in the response to the reviewer's previous point, we agree that the presentation and purpose of figures S6-8 in the supplement was unclear. We have streamlined the supplemental figures to be less redundant and further present here (Fig. R7) a selection of maps from the 15 nm film that are intended to show the larger scale structure of polarization domains.

The nanoscale polarization domains seen in these films are not unique to thin films of TbInO_3 . They have also been characterized in thin films of isostructural materials. We have added additional discussion and references to the text as follows:

The terbium sub-lattice further displays the “up-up-down” displacement pattern which is a consequence of the trimerization of the lattice structure (Fig. 2(a)). This lattice distortion drives the ferroelectric order in hexagonal indates, just as in hexagonal manganites [14, 15] and ferrites [22]. By fitting the corrugation of the terbium lattice, both the amplitude and phase of the order parameter can be spatially resolved [23]. The trimerization mapping shown in Fig. 2(b-d) reveals

FIG. R6. Failed attempts to map polarization domains with strain maps. (a) A HAADF-STEM micrograph of an example region which shows no clear contrast difference between domains. (b) Local mapping of the polarization domain structure highlighting the presence of a ‘down’ domain within an otherwise largely ‘up’ polarized field of view. (b) The phase of a sinusoid fit to the local trimerization distortion further showing the domain structure. (d-f) Maps of the phase of the distortion extracted by selecting various superlattice peaks in the fourier transform which each demonstrate no clear connection to the true domain structure mapped based on atomic positions.

a multi-domain state with a preference of up-polarized domains at the film-substrate interface, interspersed with down-polarized domains that tend to widen towards the top surface of the film. This is in contrast to the preferably single-domain state found in both hexagonal manganites and ferrites grown on the same type of YSZ substrate [24, 25]. A similar nanometer-scale multi-domain state is realized in \$\text{YMnO}_3\$ thin films on platinum-coated \$\text{Al}_2\text{O}_3\$ substrates [26,27]]. The TbInO_3 thin films display all six trimerization domains (Fig. 2(b-d), Figs. S7-S8) with a smaller domain size and thus larger number of domain walls in comparison

FIG. R7. Polar distortion amplitude maps of multiple regions (a-c) of a 15 nm TbInO_3 thin film.

to the bulk crystals of TbInO_3 [10]. A thinner 8 nm TbInO_3 also shows a similar domain pattern (Supp. Fig. S7). It has been proposed that the domain walls could host magnetic edge states or novel spin excitations [10], making the TbInO_3 thin films with rich domain structure [23] an exciting platform to study these emergent properties.

In Fig S5a, many local contrast differences can be found in the HAADF-STEM of TbInO_3 thin films. Could we assume it comes from chemical variations (Z contrast)? Are they related to the emergence of ferroelectric domain structures?

The contrast differences seen in the large field of view TEM in Fig. S6a are caused by defects, such as antiphase boundaries, and local tilting of the film or lamella. We have imaged

these defect regions alongside the pristine regions presented in the manuscript. We found no significant variation in stoichiometry (chemical variation) within the film. We also found no correlation between the variation of contrast in the large field of view and the ferroelectric domains (Fig. R8). Considering the uniform contrast in Fig. S6b, despite the presence of multiple domains, there is no clear indication of domains by contrast alone.

Compared to the rough discussions of strain and ferroelectricity (the authors put all the obtained data in the article), the study of frustrated magnetism is very professional. They clearly showed the frustrated magnetism of TbInO₃ thin films while removing the influence of the substrate.

We thank the reviewer for their positive appraisal of our discussion regarding frustrated magnetism in these films.

FIG. R8. Atomic resolution imaging of local contrast differences. (a-d) HAADF-STEM micrographs showing a variety of regions with local variations in contrast that correspond to a variety of defects including an antiphase boundary (a), variations in z tilt (b), regions with a loss of trimerization/polarization and loss of atomic resolution possibly associated with a rotation off of the zone axis (c), and a boundary between two regions with different z tilt (d).

REVIEWER 2

In this manuscript, the authors report on the fabrication of epitaxial TbInO₃ thin films and their investigation of magnetization and non-local transport properties. Quantum spin liquids (QSLs), with their highly quantum entangled states, have generated considerable interest due to potential applications in fault-tolerant quantum computation. The fabrication of thin films of the QSL materials represents a crucial first step toward potential device applications. TbInO₃ has recently emerged as a QSL candidate material, exhibiting ferroelectricity and exotic carriers at high temperatures. However, its ground state remains elusive, particularly regarding the magnetic state of Tb1 sites. While the authors' conventional SQUID magnetization measurements show results similar to bulk crystals (albeit with some temperature scale discrepancies), authors have also employed previously unexplored experimental probes, including scanning SQUID and non-local transport experiments. Given the limited reports on the fabrication of the QSL thin film, this research direction is significant, and the TbInO₃ film growth appears to be successful. However, there are several issues regarding the analysis and discussion of the experimental results that the authors need to address. More rigorous analysis and discussion are needed to evaluate whether the paper is worthy of publication in Nature Communications.

We thank the reviewer for their careful reading of our manuscript and for the opportunity to expand on the discussion of our results, as described in the point-by-point response below.

- 1. The authors mention that the magnetic susceptibility measured by conventional SQUID magnetometry includes a contribution from the impurity within the substrate at low temperatures. The relative magnitudes of the film+substrate versus substrate-only signals need to be clarified. If the substrate signal substantially exceeds the film signal, the uncertainty in the subtracted result could be significant. Including the raw data from both measure-*

ments would enable readers to assess the uncertainty of the results. Furthermore, the methodology for subtracting the substrate background signal should be clarified. While the authors mention the "point-by-point" subtraction, they should specify whether this refers to subtraction of SQUID scan curves or subtraction of fitted magnetization values. The former approach would be preferable, but the current methodology is not adequately described.

We thank the reviewer for this suggestion. We have added a section (Supplementary Note I) to the supplement describing in detail the background subtraction used in this manuscript, which indeed relies on the subtraction of the fitted magnetization values. For comparison, we also provide the SQUID raw data (DC scan waveforms) at various measurement conditions from a film+substrate sample and the corresponding substrate-only measurement, where the substrate displays a large background relative to the total signal. As can be seen in the new Fig. S10 (R9 here), our method of subtracting the fitted magnetization values yields equivalent film-only magnetization values as the subtraction of the individual waveforms and fitting the difference signal to extract the film magnetization. We note that in the case where one data set displays a vanishing magnetization, the corresponding fitted value has a larger error and thus for those data points, subtracting the waveforms before fitting the data might be beneficial. This case corresponds to only specific temperature and field combinations and thus only affect a few data points in each data set and does not influence the determination of parameters such as the moment of the Tb ions or the Curie-Weiss temperature which are extracted from fitting a large set of data points. Indeed, we see that even when the two data sets both display large signals, subtracting the fitted values (which have a low error) is equivalent to subtracting the waveforms, given the high sensitivity of the instrument. Additionally, the agreement between the corrected SQUID magnetization data and the XMCD data (which selectively probes the Tb moment of the film), shown in Fig. S9, provides further evidence that the substrate contribution removal is correct.

2. The temperature dependence of magnetic susceptibility in the high-temperature region suggest the Tb³⁺ ion moment is 10 μ_B . The authors attribute the deviation below 5K to reduced occupation of CEF excited states

FIG. R9. Comparison of extracting the film contribution to the magnetization from subtracting the fitted magnetization and subtracting the individual SQUID DC scan waveforms.

at the Tb1 site. This interpretation is consistent with the reduced moment of 6.5 μ_B derived from the field dependence of magnetization at 1.8K. Based on this interpretation, it would be important to estimate the saturation magnetic moment of Tb3+ ions from the field dependence of magnetization above 5K, where one would expect values closer to 10 μ_B . Given the ongoing debate regarding the magnetic state at the Tb1 site, clarifying this point would be essential.

The isothermal field dependence of magnetization of thin-film TbInO₃ at both 1.8K and 15K is shown in Fig. R10 (Supplementary Figure S13). As can be seen, using a simple model considering paramagnetic behavior, we find that the extrapolated saturation magnetization at 15K is higher than what is measured at 1.8K although it does not reach the full free ion moment of Tb³⁺. A similar behavior of gradually reduced saturation moment as a function of decreasing temperature in other rare-earth based frustrated magnetic oxides has been modeled by the depopulation of thermally populated crystal electric field (CEF) levels [10]. This is likely the case also for TbInO₃, although based on this magnetization data

alone, the magnetic ground state of the respective Tb1 and Tb2 sites cannot be determined. To fully capture the magnetic state of thin film TbInO₃ at very low temperatures or at very large applied fields, a more sophisticated model taking into account the full CEF level scheme would have to be applied. These levels are usually determined from inelastic neutron scattering which is not accessible for thin film samples, and hence the CEF levels of Tb³⁺ in our films remain unknown. We have edited the discussion of the temperature dependent Tb ion magnetic state in the main text as follows:

Based on the isothermal field dependence of magnetization at 15 and 1.8 K, we extrapolate saturation moments of ca. 7.4 and 6.7 μ_B/Tb^{3+} ion, showing a gradual decrease with decreasing temperature (Fig. S13). Models of the depopulation of thermally populated crystal electric field levels in other rare-earth based frustrated magnetic oxides show a similar gradual reduction in saturation moment [29]. In previous studies on bulk TbInO₃, it has been suggested that the depopulation of the first excited crystal electric field (CEF) level on the Tb1 site (located at around 0.65 meV) leads to a singlet ground state with a much reduced or vanishing magnetic moment [9] and an effective magnetic lattice below 5 K with stuffed honeycomb geometry (Fig. 1(c)). Although our estimated saturation moment of ca. 6.5 μ_B/Tb^{3+} at 1.8 K (Fig. 3(c)) could be consistent with the stuffed honeycomb model considering a suppressed Tb1 moment [9], our data do not permit determining the exact magnetic configuration of Tb1 and Tb2 in our films. We also note that other studies of TbInO₃ bulk crystals using Raman spectroscopy [30] and inelastic neutron scattering [11], have suggested a triangular magnetic lattice with similar magnetic ground state for both Tb1 and Tb2 sites.

3. The authors present AC susceptibility measurements at dilution refrigerator temperatures using scanning SQUID that requires impressive technical capability. However, the discussion of these sophisticated experimental results requires substantial elaboration. The current manuscript does not adequately explain the implications of the observed decrease in the AC susceptibility below

FIG. R10. Isothermal magnetization curves acquired at 15 K and 1.8 K with the applied field along the ab plane of a 29-nm TbInO_3 film. The saturation moment M_S is estimated by fitting the data to the Langevin function for paramagnets.

1K. Several critical questions remain unaddressed.

We thank the reviewer for describing our capability as impressive and appreciate the opportunity to expand the discussion of our observations. We agree with the reviewer that it would be desirable to have a more precise interpretation of the observed decrease in AC susceptibility. Below we discuss some additional analysis of the data within the context of spin freezing. However, given the limited information that the measurements provide in this regard, and the lack of comparable bulk crystal data, any detailed interpretation will be quite speculative.

What types of QSL states would be consistent with these measurements? Is it appropriate to attribute the phenomena observed exclusively in AC susceptibility to dynamic effects? I would like to encourage the authors to include a more comprehensive interpretation of these experimental findings.

We have added a more comprehensive examination of our AC susceptibility measurements and the types of frustrated magnetic behavior consistent with our results in Supp. Sec. IV (and add an appropriate reference in the main text):

In this section we provide an analysis of the frequency dependence of the AC susceptibility data presented in Fig. 3(d). In many experimental manifestations of a QSL state, the spins gradually lose their ability to reorient freely at the lowest temperatures; instead, they only rotate over much longer timescales that increase with decreasing temperature. Measurements on the bulk crystal TbInO₃ show that although the spin fluctuations slow down with decreasing temperature, the spins remain in a fluctuating state down to the lowest temperatures (100 mK) measured by μ SR [2]. In the case where spin freezing occurs, it is typically accompanied by a broad turnover in the AC susceptibility as a function of temperature, and the temperature at which the AC susceptibility has a maximum is often referred to as the freezing temperature. This freezing temperature is frequency dependent, see e.g. [5-7], and roughly occurs when the experimental frequency becomes comparable to the characteristic relaxation rates in the system.

We observe a frequency dependence of the AC susceptibility and the temperature at which the turnover occurs. In Fig. R11(a) we analyze the temperature dependence of the freezing temperature by fitting a polynomial to the AC susceptibility to interpolate between the data points and extract the position of the maximum, T_m , as a function of frequency shown in Fig. R11(b). Following Refs. [5,6,8], we fit the frequency dependence of T_m to a power law expression:

$$\tau = \tau^* \left(\frac{T_m}{T_g} - 1 \right)^{-z\nu} \quad (\text{R1})$$

where $\tau = (2\pi f)^{-1}$, τ^* is the characteristic relaxation time, T_g is the static spin glass freezing temperature, and $z\nu$ is the critical exponent. The fit to our extracted T_m is shown in Fig. R11(c) and shows good agreement with a power law scaling of T_m . We find the fit parameters to be: $\tau^* \sim 2.0 \times 10^{-5}$ s, $T_g \sim 0.63$ K, and $z\nu \sim 11.3$. An additional analysis [5,6,8,9] is extracting the temperature shift of the freezing temperature ΔT_m per decade change in frequency:

$$K = \frac{1}{T_m} \frac{\Delta T_m}{\Delta \log_{10} \omega} \quad (\text{R2})$$

FIG. R11. (a) AC susceptibility as a function of temperature and frequency. The data at each frequency are fit to a polynomial near the turnover (black lines) and the temperature T_m is determined as the temperature at which the polynomial function has a maximum (red points). The y-axis corresponds to the 7 Hz dataset, all other curves have been offset for clarity. (b) Frequency dependence of the freezing temperature T_m , corresponding to the red points in (a). (c) Frequency dependence of the freezing temperature with the line representing the best power law fit to Eq. R1. (d) Temperature shift of the freezing temperature as a function of changes in frequency. The line corresponds to the best fit to Eq. R2.

We extract K for our data by fitting to Eq. R2 as shown in Fig. R11(d) and find a value of $K \sim 0.076$. The values of τ^* and K obtained from our analysis are larger than than those of a canonical spin glass: $\tau^* \sim 10^{-12} - 10^{-13}$ and $K \sim 10^{-3}$. Our parameters fall closer to, although do not coincide with, those expected for a cluster spin glass ($\tau^* \sim 10^{-7} - 10^{-10}$ and $K \sim 10^{-2}$) in which local groups of spins are ordered into domains, but each domain fluctuates in a

similar way to individual spins in a spin glass [7-10]. We note, however, that the extracted parameters of τ^* and K are sensitive to the extraction method for T_m e.g. the order and temperature range of the polynomial fit, so precise classification of our TbInO₃ sample through considering the values of τ^* and K is fairly speculative.

Spin freezing does occur in geometrically frustrated magnetic systems, but in principle some disorder in the antiferromagnetic exchange needs to be present. While the thin film shows a high level of crystallinity, a possible source of disorder in addition to defects can be ferroelectric domains. We note that a downturn in the AC susceptibility has been recently observed in bulk crystals of the isostructural hexagonal rare-earth indiate DyInO₃ [11], and has been interpreted as a signature of spin freezing with ferroelectric domains as a possible source for disorder. Taken together, our measurements suggest that spin freezing occurs at low temperature, however, given the many different mechanisms that can lead to a downturn in the AC susceptibility, it is challenging to truly pinpoint its origin.

What mechanisms could explain the discrepancy between AC and DC susceptibility temperature dependencies below 1K?

First, we note that the DC susceptibility along the c-axis, which is more directly related to the AC magnetic susceptibility was measured down to 1.8 K. The DC measurements extending down to 400 mK were performed along the ab-plane. The discrepancy between the AC and DC susceptibility data may therefore stem from the system's magnetic anisotropy, i.e. only spin fluctuations out-of-plane start to slow. In addition, there is a difference in the applied probe field applied during the measurements. The DC susceptibility was measured with an applied field of 200 mT to achieve sufficient signal, whereas in the AC susceptibility measurement, a local field of $\sim 500 \mu\text{T}$ is applied to the sample. In some magnetic systems, such as glass-like systems, the turnover in magnetic susceptibility can be sensitive to the applied field, with larger fields potentially suppressing the turnover [11, 12]. Therefore, the different field scales could contribute to discrepancies in the AC and DC susceptibilities.

Finally, as noted by the reviewer, the relevant timescales for the AC and DC measure-

ments are significantly different. The AC susceptibility measurements were carried out at fixed frequencies with the lowest value of 7 Hz, and hence a characteristic time scale of ~ 100 ms. The total “observation time” of the DC measurement is on the order of one minute, so the characteristic timescale is significantly longer. Roughly, the turnover in the susceptibility is observed when the experimental timescale becomes comparable to the characteristic relaxation timescales in the system, and therefore shifts to lower temperature as the observation time of the experiment becomes longer. This could push a possible turnover in the DC susceptibility to significantly lower temperatures than an AC susceptibility measurement at even relatively low frequencies of a few Hz. If we extrapolate the observed frequency dependence using the analysis presented in Fig. R11, we would expect a turnover in a DC susceptibility measurement to occur around 0.63 K, compared to the turnover observed at ~ 0.97 K for our AC susceptibility measurement at 7 Hz. Although in principle we would observe a turnover in our DC susceptibility around 0.63 K, we emphasize again that the analysis in Fig. R11 carries significant uncertainty, and the magnetic anisotropy and/or the different field scales used in the DC and AC measurements could further shift a possible turnover in the in-plane DC susceptibility in temperature.

We have added a condensed version of the above discussion to the manuscript by expanding and rewriting the paragraph in which we discuss the AC susceptibility data. We hope that the reviewer finds the discussion improved:

We next assess the AC susceptibility of the TbInO_3 film, using scanning SQUID microscopy, in the same regime below 5 K where the higher energy CEF levels are depopulated and TbInO_3 likely exists in its magnetic ground state. To characterize the substrate contributions to the thin-film susceptibility, we ion-mill etch away part of the film, exposing the bare substrate underneath (see Methods and Supp. Note III for details). We use a scanning SQUID microscope in a dilution refrigerator to locally measure, with micrometer scale spatial resolution, side-by-side the susceptibility of the bare substrate and film + substrate together. The probe field is generated by an excitation coil concentric with the SQUID’s sensitive area, which applies a local field of approximately \$500 \mu\text{T}\$ along the \$c\$ -axis of the sample. This approach therefore probes the out-of-plane susceptibility of the sample. In the weak screening limit, the susceptibility of the film is found

by subtracting the bare substrate susceptibility from the susceptibility measured from the film + substrate. This allows us to determine the magnetic susceptibility specific to the TbInO_3 film down to 44mK, an order of magnitude lower in temperature than previous work [9].

Figure 3(d) shows the AC susceptibility measured with the field along the c -axis of TbInO_3 from 44mK to 4K. At higher temperatures, the AC and DC susceptibilities exhibit similar behavior. However, in the AC susceptibility a maximum is observed around 1K, below which the susceptibility decreases. In addition, the temperature at which the maximum susceptibility occurs weakly depends on the applied AC frequency. The AC susceptibility appears spatially uniform within our spatial resolution.

A downturn in the AC magnetic susceptibility is observed in various magnetic systems - including spin glasses, two-dimensional spin liquid candidates, superparamagnets, systems that magnetically order and combinations of these. In each system different mechanisms are at play to cause a downturn in the AC susceptibility [31-35]. Our frequency dependence is consistent with reports for spin glasses in which disorder and geometric frustration hinder long range order [31] (see Supp. Sec. IV for an analysis of the frequency dependence). In spin glasses, the downturn occurs at a temperature at which the relaxation rates fall below the excitation frequency.

Our in-plane DC susceptibility shows paramagnetic behavior down to at least 400 mK (Fig. 3(b)) without a turnover as seen in the AC susceptibility. Generally, a turnover in DC susceptibility would occur at a lower temperature than in AC measurements. Extrapolating our frequency dependence using models for spin freezing transitions (see Supp. Sec. IV), a turnover in DC would be expected around 600mK. However, the DC and AC susceptibility measurements probe along different crystallographic directions in this anisotropic system raising the possibility that spin fluctuations are not freezing in the in-plane directions. In addition, the applied probe field in the DC measurements is substantially larger than the AC field, which may result in a temperature shift of a turnover in the

in-plane DC susceptibility [36,37].

4. Following from the previous point, the authors suggest that the disorder effects cannot be ruled out as the origin of the decrease of the AC susceptibility, but this requires further clarification. What specific possibilities are they considering? Given that scanning probe measurements were employed, couldn't spatial dependence data be used to evaluate the potential disorder effects?

Spin glasses and spin freezing are often associated with disorder and disorder has been studied as a way to tune between spin liquid-like ground states to spin glass-like ground states [12]. Although the TbInO₃ thin films exhibit a high degree of crystallinity and XRD and EELS data do not suggest significant disorder from stoichiometry or impurities, certain kinds of disorder may still be present that can influence the spin coupling in the system. For example, the varying strain profiles at the sample-substrate interface (Fig. R1 and R2) or at the ferroelectric domain boundaries could modulate the spin correlations and result in some amount of short range interaction. Our scanning SQUID probe averages over an area of $\sim 5 \mu\text{m}^2$, and has a lateral spatial resolution on the micron scale. We are therefore not sensitive to atomic scale disorder. On a micron length scale, the magnetic response of the TbInO₃ films appears uniform across the sample (see Fig. R12, new Supp. Fig. S18).

5. The authors attempt to correlate their non-local transport signals with exotic carriers previously observed in the THz measurements. However, there is a significant discrepancy between these observations: while THz measurements reported temperature-independent exotic carriers between 150K-300K, the present study shows that the R1 signals related to SHE emerge only above 300K. These two observations cannot be straightforwardly reconciled, suggesting a gap in the current discussion. A more thorough discussion is warranted on the origin of the observed non-local transport. In particular, the observation of magnetism-related transport phenomena at temperatures more than an order of magnitude above the Curie-Weiss temperature challenges conventional understanding. While a complete explanation may be challenging, the authors

FIG. R12. Spatial image of the AC SQUID susceptibility (corresponding to ΔM as described in Supp. Sec.II) taken at the boundary of the TbInO₃ film and where it has been ion-mill etched away revealing the bare YSZ substrate underneath. The scanning SQUID has μm scale spatial resolution, and the TbInO₃ appears uniform on these length scales.

should explore several possible scenarios to address this unusual behavior.

Our results reveal possible thermally activated transport carriers near room temperature, corresponding to an energy of a few tens of meV (26 meV at 300 K). Similarly, the THz study probes a comparable energy range (62 meV at 50 cm^{-1}). Although the origin of these signals is not yet fully understood, both measurements suggest the possible presence of high-temperature carriers hosted by TbInO₃. As pointed out by the referee’s comment, spin-exchange interactions cannot drive the high temperature transport phenomena we report. One possible mechanism more in line with the high temperature scale is through phonons. Several recent works explore the role of acoustic phonons [13–15] and optical phonons [16, 17] in transport phenomena, either through magnetoelastic couplings or direct couplings to local moments. While the precise role of phonons in the transport in TbInO₃ is still under investigation, they provide natural starting points for transport in the room temperature energy scale.

6. Regarding the non-local transport data, the authors present only the symmetric component. What is the behavior of the raw data? From the current presentation, it is impossible to assess whether the symmetric part attributed to SHE is sufficiently large enough to be accurately measured compared to the raw data. The authors should provide a more comprehensive presentation of their transport measurements to substantiate their conclusions.

Fig. R13 shows the unprocessed data for both positive and negative I_{inj} . The signal exhibits a large symmetric component as the temperature is increased.

FIG. R13. Raw data from the non-local measurements of a device with 400 nm channel separation as a function of temperature.

Overall, while the successful fabrication of the epitaxial thin films of the QSL candidate material represents a significant achievement and the thin film preparation enables novel measurements such as non-local transport, the discussion and analysis of these results are insufficient. I cannot recommend the manuscript for publication in Nature Communications in its current form. If the authors wish to emphasize the importance of thin film fabrication, they need to provide more elaborate discussions and comprehensive analyses of the experimental results that have become accessible through thin film preparation.

We appreciate the reviewer's recognition of our achievement in synthesizing one of the

first thin films of a QSL candidate material. As outlined in the response to the points raised above, we have elaborated on the discussion of our results and additionally added further discussion on the importance and corresponding implications of establishing thin-film fabrication of QSL systems.

REVIEWER 3

This paper by Nordlander et al. demonstrates the growth of epitaxial thin films of TbInO₃ on YSZ substrates. Authors have put exceptionally great efforts into optimizing the growth conditions to produce phase pure films of TbInO₃, as confirmed via in-situ RHEED, XRD, AFM, and STEM results. The presence of ferroelectric domains in the films has been shown on multiple samples, and the results are in close agreement with the bulk counterparts. The absence of magnetic order down to at least 0.4 K (despite a Curie-Weiss temperature of -11K) is inferred using AC and DC -susceptibility (via scanning SQUID) along different crystalline orientations and XMCD measurements, which gives signatures of promising quantum spin liquid state in these films. Additionally, unconventional transport in these films using Pt/TbInO₃ heterostructures is also demonstrated, indicating the presence of unusual charge dynamics in this system near and above room temperatures (well above Curie-Weiss temperature). Such observations make this system an exciting candidate for exploring physical phenomena beyond the low-temperature regime. On top of all, these results were backed up with theoretical calculations performed using DFT + U + SOC along with DFPT calculation.

The paper is well-written and explores interesting areas of physics, combining thin film magnetism (quantum spin liquids in particular, which is very uncommon and sparingly reported), ferroelectricity, and non-local transport, which meets the interest of the broad audience of Nature Communication journal.

We thank the reviewer for their appreciation of our work and for providing valuable feedback on our manuscript. We address the reviewer's suggestions below.

I have the following comments/questions for the author, which need ex-

planation. 1. The title “Signatures of Quantum Spin Liquid Ground State in Epitaxial Thin Films of TbInO₃” seems very specific, demonstrating a particular property, which, in my view, can be made general to involve the other results discussed in the manuscript. (Suggestion).

We thank the reviewer for this suggestions. To reflect the broader scope of the manuscript, we have changed the title to:

“Signatures of quantum spin liquid state and unconventional transport in thin film TbInO₃”

2. The ferroelectric domain (using STEM) of many different films has been shown in the main text and supplemental information (SI). I found that all of these are nearly 15 nm in thickness. It would be interesting to know how these domains evolve with the thickness of the film.

Figures 2 and S6-8 presented domain mapping of different regions of the same 15 nm thick sample. To explore the evolution of domain structure with film thickness, we also mapped the polarization domains in an 8 nm thick film (Fig. S7 (R14)). As elaborated in the discussion, thickness does not seem to have a marked impact on the domain structure. While the qualitative domain structure and size seems consistent across both thicknesses, collated statistics across several unique regions of each film (Fig. S8 (R15)) suggest the thinner film may have a slightly smaller average distortion magnitude. We note however that whether this effect arises due to the decrease in thickness or due to the related increase in epitaxial strain cannot be determined in our study since the two are closely coupled, i.e. the strain relaxes gradually with thickness, on our chosen substrate.

The discussion in the main text was expanded as follows:

The TbInO₃ thin films display all six trimerization domains (Fig. 2(b-d), Figs. S7-8) with a smaller domain size and thus larger number of domain walls in comparison to the bulk crystals of TbInO₃ [10]. A thinner 8 nm TbInO₃ also shows a similar domain pattern (Supp. Fig. S7).

and:

FIG. R14. Improper ferroelectric domain mapping in an 8 nm TbInO₃ epitaxial thin film. (a) A map of the local distortion amplitude overlaid on a HAADF-STEM micrograph. (b) A map of the improper ferroelectric domains in the same region as (a).

The thinner sample shows a slightly smaller displacement (Supp. Fig. S8), possibly due to strain or interface clamping imparted from the substrate. While the displacement in the thicker film corresponds to a polarization similar to the reported value of the ferroelectric polarization of the hexagonal manganites and ferrites, it is larger than the electrical polarization measured in bulk TbInO₃ [10] at 77 K, despite our slightly smaller distortion.

The polarization domain mapping of the 8 nm film and collated statistics from unique regions of each film were compiled into new supplemental figures S7 and S8 (R14 and R15).

3. The authors demonstrated the magnetic properties of TbInO₃ film with a thickness of 38 nm in Fig 3. These films appear relaxed according to the RSM data in Fig. S3 (SI). I'm wondering why there is an inconsistency in the thickness of the films between the two measurements (magnetic and ferroelectric). I would like to see the magnetization results for the thinner films (in a strained state) and the relaxed films to check how magnetism evolves with confinement to 2D.

We have acquired additional magnetization data from a 29 nm film (corresponding to

FIG. R15. Statistics of improper ferroelectric domain mapping in thin films of TbInO_3 . (a) A histogram of local distortion magnitudes extracted from 10 unique regions of an 8 nm film with a mean of 33.06 ± 0.09 pm ($N = 8810$). (b) The distribution of the magnitude and phase of the polarization of the regions considered in (a). (c) A histogram of local distortion magnitudes extracted from 7 unique regions of a 15 nm film with a mean of 34.21 ± 0.05 pm ($N = 26518$). (d) The distribution of the magnitude and phase of the polarization of the regions considered in (c).

the strain-relaxed film also shown in Fig. S3a) as well as from a 8-nm film (corresponding to the fully strained case, Fig. S3c). The 29 nm film displays a similar behavior as the 38 nm film, indicating thickness independence in this thickness regime (see Figs. S11 and S13, here R16 and R17). Although we also acquired magnetization data from the 8 nm TbInO_3 film, we emphasize that this is an incredibly challenging measurement which relies especially heavily on the careful background removal since the relative contribution from the substrate

FIG. R16. ZFC-FC susceptibility curves measured in applied fields of 1000 Oe, 500 Oe and 200 Oe along the ab plane. For all measurements, no splitting of the ZFC and FC curves is observed down to at least 1.8 K.

signal dominates over that stemming from the much reduced volume of the ultrathin TbInO_3 film (see Fig. R18a). We also note that the presence of a 2-nm thick transitional layer of a cubic phase at the substrate interface (as seen in Fig. S4a) constitutes a significant portion (about 25%) of the total film thickness. Combined, these added uncertainties in the ultrathin regime of TbInO_3 render a quantitative analysis of the magnetic susceptibility unreliable. Having noted this important caveat, we show the zero-field cooled (ZFC) and field cooled (FC) curves of the magnetic susceptibility for the 8-nm sample in Fig. R18b. The data show no signatures of phase transitions or spin freezing down to at least 1.8 K. However, the mixed structure of the film at this thickness prevents further conclusions about the TbInO_3 -specific magnetic ground state.

4. The in-plane magnetic susceptibility (χ_{ab}) shown in Fig 3(a) has significantly higher values (even at 300K) compared to other materials in the paramagnetic phase (10^{-3} to 10^{-4} emu /Oe /mol). What can be the origin of

FIG. R17. Isothermal magnetization curves at 1.8 K and 15 K for a 29-nm TbInO_3 film yield reduced Tb^{3+} saturation moments compared to the free ion value indicated by the high temperature low-field susceptibility. Using the Langevin function as a simple model for a paramagnetic system, the saturation magnetization can be estimated from the magnetization curves. The curve measured at 15 K extrapolates to a slightly higher saturation moment than the curve measured at 1.8 K. We note however that to accurately model the magnetization behavior at low temperatures or at high magnetic field in Tb^{3+} systems, the full crystal electric field (CEF) splitting of energy levels in the ground state multiplet need to be considered. Indeed, similar behavior of reduced moment at low temperatures has been observed in other rare-earth-based oxides [ref] which could be understood taking into account the temperature dependent (de-)population of CEF levels as well as the anisotropy of the system.

such high moments at room temperature?

The magnitude of the susceptibility of a paramagnetic material (which is the case for TbInO_3 at room temperature) depends on the size of the magnetic moment of its constituent ions as well as the correlation between the moments as can be modeled by the Curie-Weiss law: $\chi = C/(T - \theta_{\text{CW}})$. The Curie constant C captures the effective magnetic moment of the

FIG. R18. Temperature-dependent magnetization from SQUID magnetometry on an 8-nm sample grown on YSZ(111). The data was acquired with an applied magnetic field of 2000 Oe along the ab plane. (a) The induced magnetic moment under zero-field cooling (ZFC) and field-cooling (FC) conditions measured on the thin-film sample and the corresponding substrate measurement under the same conditions after removing the film. (b) The film-only signal is extracted by normalizing the sample data in (a) by the substrate measurement as described in Methods. Paramagnetic behavior is observed in the 8-nm film down to 1.8 K without splitting between the FC and ZFC curves.

ions, whereas the Curie-Weiss temperature θ_{CW} indicates the presence of ferromagnetic or antiferromagnetic correlations. In the case of TbInO_3 , the magnetism comes from the Tb^{3+} ions. The free ion value of the magnetic moment of Tb^{3+} is very large, ca. $9.7\mu_B$, and the paramagnetic susceptibility is proportional to the square of the effective magnetic moment. This gives rise to a large susceptibility even at room temperature, when comparing to other materials with smaller magnetic moments. As can be seen in Fig. 3(a), our susceptibility is well described by the Curie-Weiss law at 300 K given a bulk-like Tb moment and a $\theta_{CW} = -11\text{K}$. The room-temperature susceptibility measured in our thin-film sample is consistent with that measured on bulk samples of TbInO_3 and of the same order as other Tb^{3+} -based frustrated magnets such as $\text{Tb}_2\text{Ti}_2\text{O}_7$ [18, 19].

5. Also, in Fig 3(a), a tiny variation is observed in ZFC and FC curves

in the temperature range of 100-250K. Can authors comment on whether it comes from the film or has another origin?

We thank the reviewer for highlighting this aspect. The small temperature-independent offset between ZFC and FC curves is not intrinsic to the film, but may rather stem from the measurement instrument itself, as this behavior also can be seen when comparing FC and ZFC curves measured on the bare substrate (i.e. in the background measurement), see Fig. R19. Although this offset is in general quite efficiently corrected by the substrate subtraction method across many measurements, a small constant offset between ZFC and FC curves sometimes remains in the thin film data, as can be seen in Fig. 3(a) in the main text. We note that this instrument-related offset is largely temperature independent and does not affect the analysis and conclusions of the results.

6. The Methods section indicates that the ZFC-FC measurements were conducted in a field in 2000 Oe. Is it possible to check the response of the films in a lower-applied field, as it can better probe the weak spin interactions in the system?

We have now added additional ZFC-FC measurements at lower applied fields (1000 Oe, 500 Oe and 200 Oe) to the supplement (Fig. S11) as well as in Fig. R20 below. The behavior of the magnetic susceptibility at these lower applied fields is consistent with that observed at 2000 Oe. In particular, there is no splitting between the ZFC and FC curves down to at least 1.8 K.

7. Electron hopping conduction has proved to be a simple way of verifying the dimensionality of electron interaction present in the system. Have the authors tried fitting the resistivity of the films using such formalism (if it applies to these films)?

The temperature dependent resistivity of the electron hopping conduction can be described by the following equation:

FIG. R19. Temperature-dependent magnetization from SQUID magnetometry on a 29-nm sample grown on YSZ(111). The data was acquired with an applied magnetic field of 500 Oe along the ab plane. The curves correspond to the induced magnetic moment under zero-field cooling (ZFC) and field-cooling (FC) conditions measured on the thin-film sample and the corresponding substrate measurement under the same conditions after removing the film. A small offset between ZFC and FC curves is observed. Since the offset is observed also in measurements on the bare substrate, and thus is mostly removed by our substrate subtraction method, we conclude that the offset is not intrinsic to the samples but rather generated by the instrument itself.

$$\rho(T) = \frac{1}{\sigma(T)} \propto \exp\left[\left(\frac{T_0}{T}\right)^p\right]$$

Where p is related to the dimensionality d of the system $p = \frac{1}{d+1}$. In our case, the fitting to the conductance with this equation results in a poor fit (with dimensionality from 1 to 3) shown in Fig. R21, perhaps challenged by the background and relatively sparse data points at higher temperatures, leading to inconclusive results. Additionally, the extracted T_0 (Table I) yields inconsistent values where an invariance with spacing would be expected.

8. The presence of non-local transport in the heterostructures of these

FIG. R20. ZFC-FC susceptibility curves measured in applied fields of 1000 Oe, 500 Oe and 200 Oe along the ab plane. For all measurements, no splitting of the ZFC and FC curves is observed down to at least 1.8 K.

FIG. R21. Electron hopping conduction fit (dashed line) of the conductance G_{cross} of the film, with a dimensionality, d , of 1 (a), 2 (b), and 3 (c).

systems is remarkable and needs further exploration. I request the authors to emphasize more on its significance and possible use in these systems. This will strengthen the overall manuscript.

We appreciate the reviewer's positive assessment of our work. We agree that this is an exciting area for future study on QSL candidates. In response to the reviewer's comment we have updated the title of the manuscript and added additional context to the discussion.

TABLE I. The fit parameter, T_0 (K), for the electron hopping conduction for dimensions 1 to 3 and channel separations 400, 600, and 800 nm.

Dimensions	Channel Separation (nm)		
	400	600	800
1	6.19×10^4	9.66×10^4	1.10×10^5
2	2.71×10^6	5.24×10^7	6.37×10^6
3	1.67×10^8	3.99×10^8	5.19×10^8

The updated title, Signatures of Quantum Spin Liquid State and Unconventional Transport in Thin Film TbInO₃, emphasizes the importance of the nonlocal spin transport measurements.

Strikingly, enabled by our thin film samples, nonlocal transport measurements indicate unconventional carrier transport unrelated to magnetic long-range order occurs in this system at temperatures well above $|\theta_{\text{CW}}|$ that has not been observed previously. In light of the recent report of exotic room-temperature carrier dynamics in bulk crystals of TbInO₃ probed by THz conductivity [18], our observation is further testament to the richness of exotic physics found in this system beyond the low-temperature regime of phenomena in quantum spin liquids. Future work could explore whether this exotic transport is unique to the spin liquid candidate TbInO₃ or more generic to other highly-entangled magnets.

9. What are the author's thoughts about the effect of dimensional confinement (in thin film geometry) on the TbInO₃ system? Are there changes in properties that distinguish them from their bulk equivalents?

We find that many aspects of the material are similar between the bulk crystals and the thin films: we find no evidence for long-range magnetic ordering in our materials to the lowest temperatures, similar to measurements on bulk crystals. We also find the same ferroelectric Tb-ion distortion in our materials, although our domain structure imaged differs from that of the bulk crystals. We do not find additional variation in the ferroelectric domain pattern

when imaging our 8 nm thick film in comparison to the 15 nm thick film, suggesting that there is not an additional impact of dimensionality once in the thin film limit. We have extended the conversation in the main text regarding the ferroelectric domains and Tb-ion displacement. This is also addressed in the response to inquiry 2.

Our manuscript reports a number of new findings on TbInO₃ materials including the AC susceptibility data and the new spin transport, uniquely enabled by the thin film geometry. As these measurements have not been performed on the bulk crystals, we cannot comment at this stage whether they are unique properties of our thin film samples. We find this to be a very exciting avenue for study in both the bulk samples of TbInO₃ as well as other thin films of quantum spin liquid candidates. We have emphasized this future direction in the discussion in response to the prior inquiry.

-
- [1] J. Kim, X. Wang, F.-T. Huang, Y. Wang, X. Fang, X. Luo, Y. Li, M. Wu, S. Mori, D. Kwok, E. D. Mun, V. S. Zapf, and S.-W. Cheong, Spin Liquid State and Topological Structural Defects in Hexagonal TbInO₃, *Physical Review X* **9**, 031005 (2019).
 - [2] B. B. Van Aken, T. T. Palstra, A. Filippetti, and N. A. Spaldin, The origin of ferroelectricity in magnetoelectric YMnO₃, *Nature materials* **3**, 164 (2004).
 - [3] C. J. Fennie and K. M. Rabe, Ferroelectric transition in YMnO₃ from first principles, *Physical Review B* **72**, 100103 (2005).
 - [4] T. Katsufuji, M. Masaki, A. Machida, M. Moritomo, K. Kato, E. Nishibori, M. Takata, M. Sakata, K. Ohoyama, K. Kitazawa, *et al.*, Crystal structure and magnetic properties of hexagonal $r\text{mno}_3$ ($r = y, lu, \text{ and } sc$) and the effect of doping, *Physical Review B* **66**, 134434 (2002).
 - [5] T. Choi, Y. Horibe, H. Yi, Y. J. Choi, W. Wu, and S.-W. Cheong, Insulating interlocked ferroelectric and structural antiphase domain walls in multiferroic ymno₃, *Nature materials* **9**, 253 (2010).
 - [6] J. Nordlander, M. Campanini, M. D. Rossell, R. Erni, Q. N. Meier, A. Cano, N. A. Spaldin, M. Fiebig, and M. Trassin, The ultrathin limit of improper ferroelectricity, *Nature Communications* **10**, 5591 (2019).

- [7] H. Pang, F. Zhang, M. Zeng, X. Gao, M. Qin, X. Lu, J. Gao, J. Dai, and Q. Li, Preparation of epitaxial hexagonal ymno3 thin films and observation of ferroelectric vortex domains, *npj Quantum Materials* **1**, 1 (2016).
- [8] T. Matsumoto, R. Ishikawa, T. Tohei, H. Kimura, Q. Yao, H. Zhao, X. Wang, D. Chen, Z. Cheng, N. Shibata, *et al.*, Multivariate statistical characterization of charged and uncharged domain walls in multiferroic hexagonal ymno3 single crystal visualized by a spherical aberration-corrected stem, *Nano letters* **13**, 4594 (2013).
- [9] Q. Zhang, G. Tan, L. Gu, Y. Yao, C. Jin, Y. Wang, X. Duan, and R. Yu, Direct observation of multiferroic vortex domains in ymno3, *Scientific reports* **3**, 2741 (2013).
- [10] M. Gingras, B. Den Hertog, M. Faucher, J. Gardner, S. Dunsiger, L. Chang, B. Gaulin, N. Raju, and J. Greedan, Thermodynamic and single-ion properties of tb^{3+} within the collective paramagnetic-spin liquid state of the frustrated pyrochlore antiferromagnet $\text{tb}_2\text{ti}_2\text{o}_7$, *Physical Review B* **62**, 6496 (2000).
- [11] P. Bag, P. Baral, and R. Nath, Cluster spin-glass behavior and memory effect in $\text{cr}_{0.5}\text{fe}_{0.5}\text{ga}$, *Physical Review B* **98**, 144436 (2018).
- [12] R. Zhong, M. Chung, T. Kong, L. T. Nguyen, S. Lei, and R. J. Cava, Field-induced spin-liquid-like state in a magnetic honeycomb lattice, *Physical Review B* **98**, 220407 (2018).
- [13] A. Rückriegel and R. A. Duine, Long-range phonon spin transport in ferromagnet–nonmagnetic insulator heterostructures, *Physical Review Letters* **124**, 117201 (2020).
- [14] T. Uehara, T. Ohtsuki, M. Udagawa, S. Nakatsuji, and Y. Machida, Phonon thermal hall effect in a metallic spin ice, *Nature Communications* **13**, 4604 (2022).
- [15] H. Guo, Phonon thermal hall effect in a non-kramers paramagnet, *Physical Review Research* **5**, 033197 (2023).
- [16] J.-Y. Chen, S. A. Kivelson, and X.-Q. Sun, Enhanced thermal hall effect in nearly ferroelectric insulators, *Physical Review Letters* **124**, 167601 (2020).
- [17] K.-M. Kim and S. B. Chung, Phonon-mediated spin transport in quantum paraelectric metals, *npj Quantum Materials* **9**, 51 (2024).
- [18] L. Clark, G. Sala, D. D. Maharaj, M. B. Stone, K. S. Knight, M. T. F. Telling, X. Wang, X. Xu, J. Kim, Y. Li, S.-W. Cheong, and B. D. Gaulin, Two-dimensional spin liquid behaviour in the triangular-honeycomb antiferromagnet TbInO_3 , *Nature Physics* **15**, 262 (2019).

- [19] J. S. Gardner, S. R. Dunsiger, B. D. Gaulin, M. J. P. Gingras, J. E. Greedan, R. F. Kiefl, M. D. Lumsden, W. A. MacFarlane, N. P. Raju, J. E. Sonier, I. Swainson, and Z. Tun, Cooperative Paramagnetism in the Geometrically Frustrated Pyrochlore Antiferromagnet $\text{Tb}_2\text{Ti}_2\text{O}_7$, *Physical Review Letters* **82**, 1012 (1999).

Second Round Review Response for
Signatures of Quantum Spin Liquid State and Unconventional
Transport in Thin Film TbInO₃

Johanna Nordlander,* Margaret A. Anderson,* Tony Chiang, Austin Kaczmarek, Nabaraj Pokhrel, Kuntal Talit, Spencer Doyle, Edward Mercer, Christian Tzschaschel, Jun-Ho Son, Hesham El-Sherif, Charles M. Brooks, Eun-Ah Kim, Alberto de la Torre, Ismail El Baggari, Elizabeth A. Nowadnick, Katja C. Nowack, John T. Heron, and Julia A. Mundy

REVIEWER 1

This work's main contribution is demonstrating the thin-film manifestation of exotic properties, such as a potential quantum spin liquid ground state in the first TbInO₃ thin film. The authors clearly show this point with detailed data. In my opinion, the article can be published after some minor modifications. I only have some questions and suggestions about the details.

We thank the reviewer for their careful consideration of our manuscript and for their further suggestions to improve it.

1. The presented data combine results from films of varying thicknesses, such as the ferroelectric domain pattern (Fig. 2, 15 nm), SQUID magnetometry (Fig. 3, 38 nm), and isothermal magnetization curves (Fig. S13, 29 nm). Why were different thicknesses selected for those measurements?

For all samples, the crystalline quality and strain was determined by XRD as the main characterization method. STEM was additionally used on a subset of samples to correlate the XRD data with the local microstructure, with excellent agreement. STEM characterization was performed for two different film thicknesses – 8 nm and 15 nm – with nominally different strain states. Although mapping the ferroelectric domain structure (which has been extensively studied in isostructural films such as YMnO₃ or LuFeO₃) is a secondary objective to our main focus on magnetic properties in thin film TbInO₃, our data suggests minimal influence of thickness/strain on the manifestation of the ferroelectric domain pattern.

Regarding the differences in thickness for the magnetic characterization, we performed magnetometry on several samples in the thickness range 8 nm - 38 nm, all of which are fully characterized by RHEED and XRD for determining the crystalline structure and quality. Our focus in this study has been on the magnetic properties in the strain relaxed regime (29 nm and 38 nm). We emphasize that since the method we employ to ensure correct removal of magnetic background signal from the substrate is inherently destructive to the sample – it involves a step where the film is physically removed from the substrate (see

* Equal contribution

Methods) – additional thin-film characterization is not possible after this step. However, we show that similar magnetic behavior is exhibited between the 29 nm sample and the 38 nm sample, indicating robustness of properties and thickness independence at least in the strain relaxed regime.

Magnetic characterization in the strained thickness regime (corresponding to the 8 nm sample) was attempted but analysis was hindered by insufficient magnetic volume of the TbInO₃ film compared to the background magnetic contribution.

2. The authors have shown that films with varying thicknesses exhibit different misfit strains (Figs. S3-S5). However, they did not address the implications of these variations. Although the films with different thicknesses display similar ferroelectric domain structures, other possible differences – such as magnetic properties – were not discussed.

We note that although the main focus of this work is the magnetic properties of thin-film TbInO₃ in the strain-relaxed regime, we did perform SQUID magnetometry for samples in both the strained (8 nm) and strain-relaxed (29 nm, 38 nm) films as discussed above and in the previous response letter (Question 1 of Reviewer 1 and Question 3 of Reviewer 3). However, since the complete and unambiguous magnetic characterization was precluded in the strained 8-nm film due to small sample volume, we have opted here not to speculate about the magnetic properties in this low-thickness regime. The strain dependence of magnetic properties in TbInO₃ remains a topic of future work, as such analysis would require different measurement capabilities and falls outside the scope of the present study. We hope that our results presented here will inspire and enable such additional follow-up studies in the future.

3. The authors mention a cubic transitional layer in TbInO₃ films. Figs. S4/S5 appear compressed during editing, making identifying the transitional layer from STEM images difficult. Based on the strain maps, 8 nm film has a cubic transitional layer (strain $\epsilon_{yx} \sim 0$), but the transitional region in 15 nm film appears non-cubic. Additionally, I suggest adding a brief discussion about the transition layer.

We reproduce here figures S4 and S5 in full resolution (Figs. R1 and R2), including

FIG. R1. Strain mapping in an 8 nm TbInO₃ on YSZ(111). (a) A HAADF-STEM micrograph of the 8 nm thick TbInO₃ film (top) and YSZ substrate (bottom) and a transitional layer with a cubic structure in between. (b-c) Strain maps of ϵ_{xx} and ϵ_{yx} in the region shown in (a) derived from a lock-in phase analysis analogous to geometric phase analysis. Uniform ϵ_{xx} across the interface (b) suggests the film is fully strained to the substrate. Finite ϵ_{yx} clearly identifies regions with the ideal hexagonal TbInO₃ structure. **The cubic transition layer is identified as the interface region with similar STEM contrast as the TbInO₃ film in (a) yet with $\epsilon_{yx} \sim 0$ in (c), and corresponds here to the first four atomic layers away from the substrate interface. The cubic phase of this transitional layer is likely seeded by the cubic symmetry of the substrate.**

revised captions adding additional details about the transition layer.

We agree with the reviewer that the transitional layer is especially difficult to identify in the 15 nm film strain map. We next annotate the above figures to show the substrate-film interface and the transition of ϵ_{xy} to nonzero values (Figs. R3 and R4). Possibly due to slight differences in the resolution or scale of the original image used for Fig. S5 (R2), the boundary with nonzero ϵ_{xy} is less sharp. Nonetheless, the annotated images show that the nonzero shear strain arises one terbium ion monolayer below the visible appearance of the trimerizing distortion in the TbInO₃ film, a few monolayers above the substrate-film interface, in both cases. The cubic transitional layer may be thinner in the 15 nm film due to the partial strain relaxation within the film as identified in Fig. R2.

Because the transitional layer has negligible impact on film properties, we limit discussion of the transitional layer to the supplement.

FIG. R2. Strain mapping in 15 nm TbInO₃ on YSZ(111). (a) A HAADF-STEM micrograph of the interface between the TbInO₃ film (top) and YSZ(111) substrate (bottom) with a cubic **transitional region of about three atomic layers at the interface determined in the same way as in Fig. S4**. (b-c) Strain maps derived from a lock-in phase analysis showing increased ϵ_{xx} at the interface and small negative ϵ_{xx} within the film (b), which indicates partial relaxation of the film, and homogeneous, finite ϵ_{yx} within the film (c).

FIG. R3. Annotated strain mapping in 8 nm TbInO₃ film. (a-c) added red (blue) dashed line identifying the substrate-film interface (onset of nonzero ϵ_{yx})

4. Please provide the technical details of strain mapping in Methods.

We thank the reviewer for identifying the accidental omission of the strain mapping methods.

Our strain maps used the lock-in technique developed and described in Goodge, Berit H., et al. "Disentangling coexisting structural order through phase lock-in analysis of atomic-resolution STEM data." *Microscopy and Microanalysis* 28.2 (2022): 404-411.

FIG. R4. Annotated strain mapping in 15 nm TbInO₃ film. (a-c) added red (blue) dashed line identifying the substrate-film interface (onset of nonzero ϵ_{yx})

We add the following statement to the STEM methods (the full STEM methods paragraph is quoted in response to the reviewer’s point 5 below):

Local strain variations were mapped using the phase lock-in technique developed in Ref. [41].

5. Line 339. “...using an FEI Helios 660 Focused Ion Beam (FIB) with a final milling step of 2 keV to reduce surface damage.” I think that 2 keV should be 2 kV.

We appreciate the reviewer’s close reading of the methods and for their suggestions to improve the clarity of the text.

Because this description of the FIB milling process is referring to the energy of the ions, keV is the appropriate unit of measure in this case. We amend the language of the selected phrase for clarity:

Cross-sectional scanning transmission electron microscopy (STEM) specimens were prepared using an FEI Helios 660 Focused Ion Beam (FIB) with a final milling step at a beam energy of 2 keV to reduce surface damage.

For consistency, we also amend the description of the imaging accelerating voltage to give the beam energy as well. The full STEM methods section now reads:

Cross-sectional scanning transmission electron microscopy (STEM) specimens were prepared using an FEI Helios 660 Focused Ion Beam (FIB) with a final milling step at a beam energy of 2 keV to reduce surface damage. High-angle annular dark-field STEM measurements were performed either on a JEOL ARM 200F or a Thermo-Fisher Scientific Titan Themis Z G3 both operating at 200 keV beam energy. The convergence angle was either 19.6 or 22 mrad and the collection angle range was ~ 68 mrad-280 mrad. Local strain variations were mapped using the phase lock-in technique developed in Ref. [41]. For mapping the improper ferroelectric domains and trimerization, atomic column positions were located by 2D gaussian fitting. Trimerization amplitude was defined as the difference in the average [001] displacement of the ‘up’ and ‘down’ terbium atoms. The average displacement amplitude in the film was calculated from averaging over $N \sim 22500$ atomic positions. The phase of polarization domains was mapped based on fitting to a sinusoid using the method detailed by Holtz et al [23].

REVIEWER 2

In the revised manuscript, the authors have expanded the discussion of their experimental results. Although some aspects remain unclear, it is often challenging to fully resolve all issues in strongly correlated systems, particularly in quantum spin liquids (QSLs). Given the significance of the subject and the scarcity of reports on the fabrication of QSLs, I believe that the current manuscript meets the standards for publication in Nature Communications.

We thank the reviewer for their positive appraisal of our efforts to clarify the text and recognition of the significance of the manuscript.

REVIEWER 3

On careful examination of the revised manuscript and rebuttal, I found that the authors have worked extremely well in answering the comments, thoroughly modifying the manuscript and adding useful information to support the findings. These satisfactorily address my comments and concerns, and I hence recommend it for publication.

We thank the reviewer for their feedback and approval of the revised manuscript.